# Little evidence that Eurasian jays protect their caches by responding to cues about a conspecific's desire and visual perspective

Piero Amodio[1,2]*, Benjamin G Farrar[1,3], Christopher Krupenye[4,5], Ljerka Ostojić[1,6], Nicola S Clayton[1]

[1]Department of Psychology, University of Cambridge, Cambridge, United Kingdom; [2]Department of Biology and Evolution of Marine Organisms, Stazione Zoologica Anton Dohrn, Napoli, Italy; [3]Institute for Globally Distributed Open Research and Education (IGDORE), Sweden, Sweden; [4]Department of Psychological & Brain Sciences, Johns Hopkins University, Baltimore, United States; [5]Department of Psychology, Durham University, Durham, United Kingdom; [6]Department of Psychology, Faculty of Humanities and Social Sciences, University of Rijeka, Rijeka, Croatia

*For correspondence:
piero.amodio@cantab.net

Competing interest: The authors declare that no competing interests exist.

**ABSTRACT** Eurasian jays have been reported to protect their caches by responding to cues about either the visual perspective or current desire of an observing conspecific, similarly to other corvids. Here, we used established paradigms to test whether these birds can – like humans – integrate multiple cues about different mental states and perform an optimal response accordingly. Across five experiments, which also include replications of previous work, we found little evidence that our jays adjusted their caching behaviour in line with the visual perspective and current desire of another agent, neither by integrating these social cues nor by responding to only one type of cue independently. These results raise questions about the reliability of the previously reported effects and highlight several key issues affecting reliability in comparative cognition research.

## Introduction

Theory of mind is thought to provide a causal and flexible cognitive framework that allows humans to navigate complex social interactions (*FeldmanHall and Shenhav, 2019*; *Tamir and Thornton, 2018*). Through this framework, different observable social cues can be used to infer otherwise imperceptible mental states (e.g., perspectives, desires, knowledge, beliefs), such that the behaviour of other individuals can be interpreted, predicted, and manipulated based on an interplay of different mental states (*Baker et al., 2017*; *Bartsch and Wellman, 1989*; *Bennett and Galpert, 1992*). In humans, theory of mind is thought to emerge as a stepwise process, with a meta-representational framework in place by at least 5 of age (*Wellman, 2018*; *Wellman and Liu, 2004*). However, even before they develop this meta-representational theory of mind, infants can already respond to multiple pieces of social information in an integrated manner (e.g., *Moses et al., 2001*; *Repacholi et al., 2014*). Multiple accounts exist, arguing that they may do so with or without representing others' mental states as such (*Apperly and Butterfill, 2009*; *Carruthers, 2020*; *Scott and Baillargeon, 2017*; *Southgate, 2019*). At a minimum, though, infants seem to implicitly register social cues and exhibit adaptive responses accordingly (*Apperly and Butterfill, 2009*; *Butterfill and Apperly, 2013*). Consequently, different mechanisms may allow individuals to integrate information from multiple social cues.

**eLife digest** Eurasian jays, *Garrulus glandarius*, are members of the crow family. These large-brained birds hide food when it is abundant, and eat it later, when it is scarce. Previous studies have found that jays avoid theft by other jays by carefully deciding what food to hide, and where. In one study, they preferred to hide their food behind an opaque barrier, rather than a transparent one, when another jay was watching. In a second study, they preferred to hide food that the watching jay had already eaten enough of, and thus did not want.

These studies suggest that jays have flexible cognitive skills when it comes to protecting their food. They respond to whether a potential thief can see their hiding place and to how much a thief might want the food they are stashing. The next question is, can Eurasian jays combine these two pieces of information? For example, if a jay has two types of food they could hide when another jay is present, but only has one place to hide them (either in view or out-of-view of the other jay), does the first jay prefer to stash the food that the second jay has already eaten, and therefore does not want anymore, only when the hiding place is visible to second jay?

To find out, Amodio et al. watched Eurasian jays hiding macadamia nuts or peanuts in the presence of another jay. In the first setup, jays were given one food to hide and two possible hiding places, one opaque and one transparent, while being watched by a jay that had either had its fill of the food, or not tried it. In the second setup, jays were given both foods to hide, but only had one place to hide them (either transparent or opaque); while being watched by a jay that had eaten enough of one of the foods.

Contrary to expectations, the jays did not seem to be able to combine the information about what the other jay could see and what it had been eating. In fact, they seemed unable to respond to either piece of information. When Amodio et al. repeated the original experiments, the jays did not seem to prefer to hide food out of sight, or to hide food that the watcher had already eaten.

These results raise questions about the repeatability of experiments on food hiding strategies in birds of the crow family. It suggests that previous findings should be further investigated, potentially to identify important factors that might affect the repeatability of food-hiding tactics. Repeating the experiments may show how best to investigate behavioural patterns in jays in the future.

Inspired by *Premack and Woodruff, 1978*, the past few decades have seen growing efforts to investigate whether non-human animals also possess something akin to theory of mind (e.g., primates: *Buttelmann et al., 2007*; *Drayton and Santos, 2014*; *Flombaum and Santos, 2005*; *Hare et al., 2000*; *Hare et al., 2001*; *Kano et al., 2019*; *Krupenye and Call, 2019*; dogs: *Horowitz, 2011*; *Maginnity and Grace, 2014*; corvids: *Bugnyar et al., 2016*; *Dally et al., 2006*; *Emery and Clayton, 2001*; *Ostojić et al., 2013*; *Shaw and Clayton, 2012*). This issue is significant, given foundational debates in cognitive science about whether minds can represent mental state concepts in the absence of language as well as a long-standing interest in the evolution of the key cognitive traits that make us human (e.g., *Penn et al., 2008*). However, within this line of research, studies most often focus on testing whether a given species/group has the ability to attribute one specific type of mental state, that is, exclusively focusing on belief, or exclusively focusing on desire, within a single study. As a result, very little is known about whether other species can – like humans (e.g., *Baker et al., 2017*) – integrate multiple social cues that correlate with others' mental states and exhibit appropriate responses accordingly. This question is far from trivial: in real life scenarios, an individual's behaviour is likely to result from the interplay of multiple factors (e.g., their perspective, desires, and previous knowledge) that can be indirectly perceived simultaneously by another agent during social interactions. Therefore, by focusing on non-human animals' ability to respond to a single social cue at a time, comparative psychologists may overlook a crucial aspect of social cognitive complexity. Additionally, investigating whether and how other species may integrate different social cues can help shift the focus of social cognition studies in animals away from binary questions – for example does species X understand false beliefs? – towards more process-oriented and nuanced research questions, such as: what are the relative contributions of mechanisms A and B to how species X perform behaviour Y? (*Buckner, 2013*; *Heyes, 2015*). In doing so, this work may also identify mechanisms of varying complexity that operate in the absence of language and may feed into uniquely human social cognition.

Corvids are a group of large-brained birds that are hypothesised to have evolved sophisticated cognitive abilities independently from primates (*Clayton and Emery, 2015*; *Emery and Clayton, 2004*; *Güntürkün and Bugnyar, 2016*; *Osvath et al., 2014*; *Seed et al., 2009*). This group represents a good model for this line of research because they might be capable of responding – independently – to social cues correlating with different types of mental states (perspective: *Bugnyar et al., 2016*; *Dally et al., 2004*; *Dally et al., 2005*; *Legg et al., 2016*; *Legg and Clayton, 2014*; *Shaw and Clayton, 2013*; *Stulp et al., 2009*; desires: *Ostojić et al., 2013*, *Ostojić et al., 2016*, *Ostojić et al., 2017Apperly and Butterfill, 2009*; knowledge: *Bugnyar and Heinrich, 2005*; *Dally et al., 2006*; *Emery and Clayton, 2001*). In particular, previous research has reported that Eurasian jays (*Garrulus glandarius*) may be able to adjust their behaviour according to cues that correlate with the perspective and current desire of a conspecific. In the presence of a conspecific competitor, these jays have been reported to preferentially cache food in less visible locations (e.g., behind barriers, at distance), or in non-noisy substrates, which has been interpreted as a potential response to the visual (*Legg et al., 2016*; *Legg and Clayton, 2014*) or acoustic perspective (*Shaw and Clayton, 2013*) of a potential pilferer. In parallel, research investigating food-sharing behaviour found that male Eurasian jays change the type of food shared with their female partner, depending on which food the female has been sated on, and therefore which she desires (*Ostojić et al., 2013*; *Ostojić et al., 2014*; *Ostojić et al., 2016*). In a recent study, a similar response to another's satiety has also been reported in the context of caching, whereby Eurasian jays and Western scrub-jays preferentially cached food that an observer, and thus potential pilferer, was sated on *Ostojić et al., 2017*. Notably, the effect reported in this study was unlikely to be based on theory of mind because the caching jays showed this effect also when they did not know what food the observer was pre-fed on and when the only cue available was the observer's behaviour during the caching event itself (*Ostojić et al., 2017*). Thus, taken together, these studies seem to indicate that Eurasian jays employ a variety of cache protection strategies to limit the risk of cache loss, by responding to cues correlating with the perspective or current desire of a potential pilferer. Although this evidence is not sufficient to demonstrate mental state attribution, nor to pinpoint the exact underlying cognitive mechanism, it does suggest that corvids are capable of behaving flexibly on the basis of different types of social cues.

The aim of the current study was to investigate whether Eurasian jays can integrate multiple cues that correlate with different types of mental states to solve social problems. Previous work has to some extent already, at least implicitly, tested whether animals can integrate perspective cues from different sensory modalities (corvids: *Shaw and Clayton, 2013*; *Stulp et al., 2009*; primates: *Bray et al., 2014*; *Santos et al., 2006*). While these studies investigate the ability to integrate different cues regarding the same type of mental state, namely another individual's perspective in assessing the significance of visual and acoustic perceptual cues, our study concerns the integration of cues correlating with different types of mental states, namely perspectives and desires. Building on previous studies in the caching context, we tested whether jays can integrate information about a conspecific's perspective and current desire to selectively protect those caches that are at most risk of being pilfered. To this end, we first conducted two experiments in which we manipulated the observer's visual access to caching locations and its current desire for different foods. We measured the caching pattern of the birds when viewed by this observer.

Our manipulation of the observer's perspective followed the procedure by *Legg and Clayton, 2014*. The authors gave Eurasian jay cachers access to two locations, one that could and one that could not be seen from an adjacent compartment and thus by an observer bird (when present). One caching tray was positioned behind an opaque barrier (*out-of-view* tray), and the other behind a transparent barrier (*in-view* tray). Legg and Clayton's experiment encompassed three conditions: jays could cache when an observer bird was housed in the adjacent compartment – the observer was either a higher ranked individual (*Observed by a dominant* condition) or a lower ranked individual (*Observed by a subordinate* condition) – or when no conspecific was present (*Private* condition). The authors compared the jays' caching pattern in the *Private* condition with that in the *Observed* conditions (the two observed conditions were merged together), and found that jays cached a higher proportion of food items in the *out-of-view* tray in the *Observed* than in the *Private* conditions.

Our manipulation of the observer's desire for different foods followed the procedure by *Ostojić et al., 2017*. The authors investigated whether Eurasian jays and California scrub-jays can protect their caches by preferentially caching the type of food that the observer was not currently motivated to

pilfer. A cacher and an observer jay were housed in adjacent testing compartments. In the pre-feeding phase, the observer could feed to satiety on a specific type of food: maintenance diet in the baseline trial, and either food A or B in the two test trials. This procedure subsequently reduces the individual's motivation for eating and caching that specific food (but not different kinds of food), a phenomenon known as 'specific satiety' (*Balleine and Dickinson, 1998*; *Clayton and Dickinson, 1999*; *Dickinson and Balleine, 1994*). *Ostojić et al., 2017* found that the jays' preference for caching food A over food B was larger after the observer was sated on food A than after the observer was sated on food B. Interestingly, this pattern was exhibited not only in the *Seen* Condition – when the cacher had witnessed which particular food had been provided to the observer – but also in the *Unseen* Condition – when the cacher had not seen the pre-feeding of the observer – thus indicating that the observer's behaviour at the time of caching may have played a key role in the decision-making process of the cacher. Note that, when data were re-analysed for Eurasian jays only, the effect was still statistically significant in the *Seen* condition – although it was not statistically significant in the *Unseen* condition (*Crosby, 2019*).

In the current study we combined these two protocols. In Experiment 1, cacher jays were provided with only one type of food – which on one trial was the same food on which the observer sated, and on another trial was a different food from that on which the observer was sated – and two caching trays, one that the observer could see and one that the observer could not see. Thus, here, the jays could choose between two caching locations, one that was in-view and one that was out-of-view of the conspecific, allowing them to selectively cache food out-of-view when it was desired by the observer. In Experiment 2, cacher jays were provided with a single caching tray – which on one trial could be seen by the observer and on another trial could not be seen by the observer – and two types of food, one of which had previously been pre-fed to the observer. Thus, here, the jays could choose between two food types, one on which the observer was sated and one on which the observer was not sated, allowing them to selectively cache the less desired food when the observer could see them. Consequently, the designs of the two experiments were complementary, such that jays could most effectively protect their caches by deciding where to cache in Experiment 1, and what to cache in Experiment 2.

## Results

### Integration of cues correlating with others' desire and perspective

In Experiment 1, we tested whether jays can integrate multiple cues to decide where to hide food to protect it from being pilfered. To do so, we manipulated the observer' visual access to two caching trays through a 'T'-shaped Perspex barrier (henceforth T-barrier). This T-barrier – which was the same barrier originally designed and used in *Legg and Clayton, 2014* – consisted of three plastic panels: one transparent panel forming one arm of the 'T' and two opaque panels forming the second arm and stem of the 'T' (see Materials and methods). The T-barrier could be placed around two caching trays (*Figure 1*) such that an observer could see the tray behind the transparent arm (*in-view* tray) but could not see the tray behind the opaque arm (*out-of-view* tray).

To ensure that the birds (n = 9) were comfortable caching in trays when these were placed in proximity of each of the two arms of the T-barrier, they initially received two familiarisation trials in private, in which only a single tray was present. The tray was placed once behind the opaque and once behind the transparent arm of the barrier. All birds except two reached the criterion in the familiarisation, that is they cached at least one item on each trial, and therefore proceeded to the test (leading to n = 7 for the test).

Following the basic design of *Ostojić et al., 2017*, the test trials comprised a pre-feeding phase and a caching phase (*Figure 1*). In the pre-feeding phase, the cacher jay could see a conspecific (the observer) eat a specific type of food (macadamia nuts, M, or peanuts, P) in an adjacent indoor compartment. In the subsequent caching phase, the cacher jay was presented with two caching trays, each placed behind one of the two arms of the T-barrier, and was allowed to cache while the observer jay was still present in the adjacent compartment. Birds were tested in two conditions (one trial per condition; *Figure 1*). In the *Different Food* Condition, the type of food received by the observer in the pre-feeding phase differed from the type of food received by the cacher in the caching phase (e.g.,

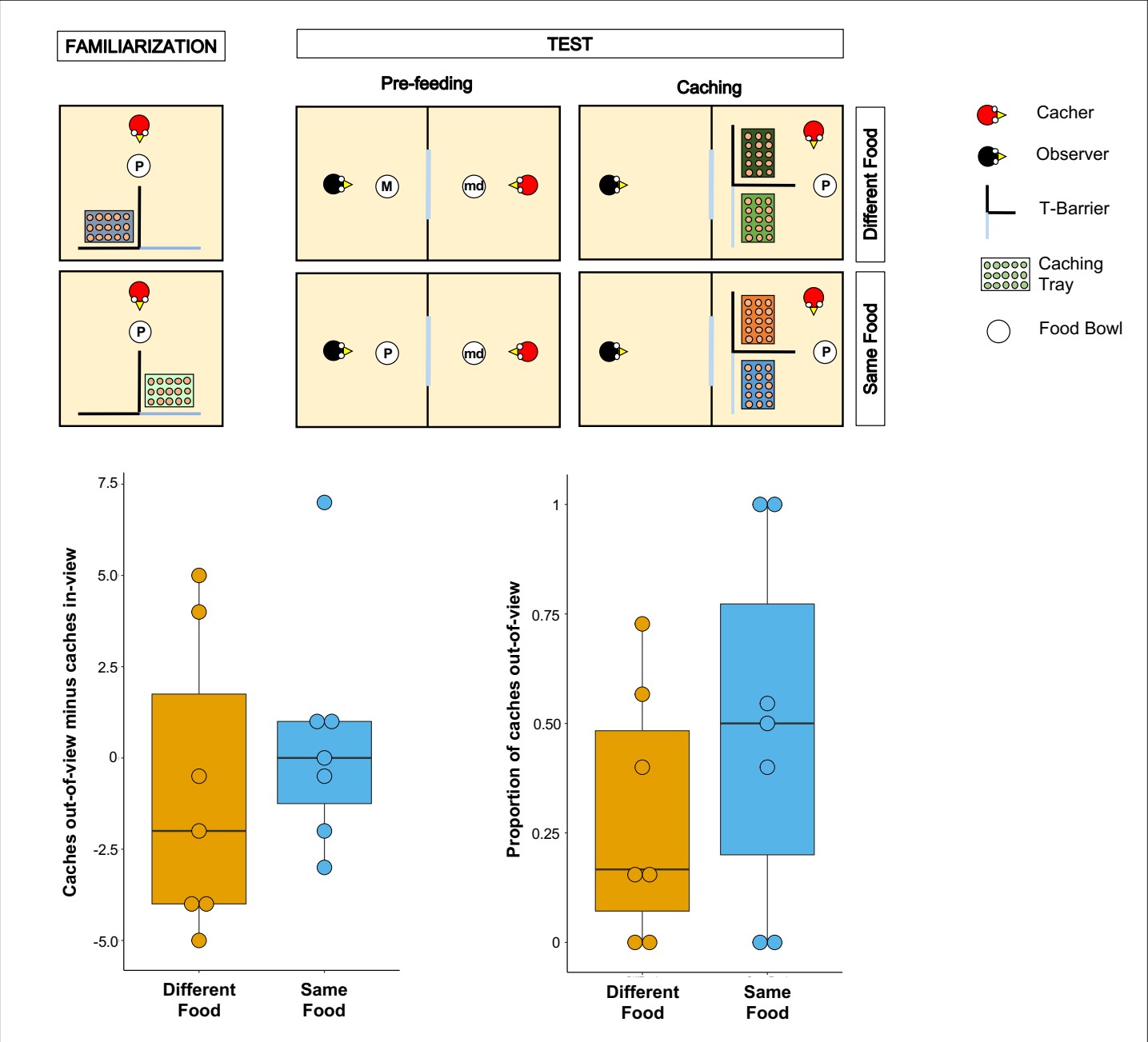

**Figure 1.** Methods and results of Experiment 1. Top panel: Top-view schematic representation of the set-up and procedure used in Experiment 1. In the familiarisation (left), the cacher bird received two trials, one in which the caching tray was placed near the opaque arm of the barrier, and one in which the tray was placed near the clear arm of the barrier. In the test, trials were composed by a pre-feeding phase (middle) and a caching phase (right). The cacher bird received two trials that differed in the type of food that was provided to the observer bird in the pre-feeding phase. In one trial (*Different Food* condition), the food provided to the observer in the pre-feeding phase differed from the food the cacher bird could subsequently cache. In the other trial (*Same Food* condition), the food provided to the observer in the pre-feeding phase was the same as the food the cacher bird could subsequently cache. In the pre-feeding phase of both trials, the cacher bird was provided with a handful of maintenance diet (md). Note that, for explanatory purposes, the scheme shows the cacher as being provided with peanuts (P) in the familiarisation and in the caching phase of the two test trials. However, in the experiment, cacher birds were randomly assigned to one type of food (either peanuts or macadamia nuts), which was used consistently in the familiarisation and in the caching phase of all trials. Bottom Panel: Box and whisker plots of data in Experiment 1. The plot on the left shows the difference in the number of items cached in the *out-of-view* tray minus the number of items cached in the *in-view* tray in the two experimental conditions. The plot on the right shows the proportion of items cached in the *out-of-view* tray (out of total caches) in the two experimental conditions.

M for the observer, and P for the cacher). In the *Same Food* Condition, the observer and the cacher received the same type of food (e.g., P for the observer, and P for the cacher).

If the jays can integrate information from the different cues correlating with the observer's desire and perspective, their caching pattern should meet two predictions. First, their preference for caching in the *out-of-view* tray should be greater in the *Different Food* condition than in the *Same Food* condition. This is because it is in the *Different Food* condition that the observer has a stronger desire toward the cacher's food, such that the caches would be more at risk from being stolen in this condition. Second, in the *Different Food* condition, the cacher should exhibit a clear preference for caching in the *out-of-view* tray, therefore in this condition the amount of caches in the *out-of-view* tray should be higher than expected by chance, namely if the cacher distributed its caches randomly across the *out-of-view* and the *in-view* trays.

The jays' preference to cache in a specific location can be indexed either as a proportion of the items cached in the *out-of-view* tray (over the items cached in both trays) or as a difference score of the number of items cached in the *out-of-view* tray minus the number of items cached in the *in-view* tray. Because robust findings would necessitate that the two different indices do not lead to drastically different results, we conducted the analyses with both indices (for details see Analysis in the Materials and methods section).

The results from the analyses with both indices were consistent. In contrast to the first prediction, a comparison of the proportion of items cached in the *out-of-view* tray between the *Different Food* and the *Same Food* conditions did not detect a statistically significant difference (Wilcoxon signed-rank test: n = 7, W = 13, *P* = 0.21; *Figure 1*). In contrast to the second prediction, the proportion of items cached in the *out-of-view* tray was not significantly different from that expected by chance in the *Different Food* condition (one-sample Wilcoxon signed-rank test: n = 7, W = 25, p=0.11). As an additional analysis, we conducted the same comparison in the *Same Food* condition and also detected no statistically significant difference from chance (one-sample Wilcoxon signed-rank test: n = 7, W = 25, p=1). The same pattern was found when the difference score – number of items cached in the *out-of-view* tray minus the number of items cached in the *in-view* tray, [Caches$_{out-of-view}$ – Caches$_{in-view}$] – was used. No statistically significant difference in the difference score was detected between the *Different Food* and the *Same Food* conditions (Wilcoxon signed-rank test: n = 7, W = 9, p=0.40; *Figure 1*) and a comparison of the difference score to chance (i.e., 0) detected no statistically significant difference in the *Different Food* condition (one-sample Wilcoxon signed-rank test: n = 7, W=-7, p=0.61). The same comparison in *Same Food* condition also detected no statistically significant difference from chance (one-sample Wilcoxon signed-rank test: n = 7, W = 1, p=1). Notably, the numerical pattern of the jays' caching was in the opposite direction to the prediction: jays' caching was more biased towards the *out-of-view* tray in the *Same Food* than the *Different Food* condition (Proportion of items cached out-of-view: Median$_{Same Food}$ = 0.5, Median$_{Different Food}$ = 0.17; Caches out-of-view minus Caches in-view: Median$_{Same Food}$ = 0, Median$_{Different Food}$ = –2). Consequently, the observed data in Experiment 1 cannot be interpreted as supporting the conclusion that the jays were integrating the information from multiple cues to protect the caches that are most at risk of being pilfered by a conspecific.

In Experiment 2, we used a complementary design to test whether jays can integrate multiple cues to decide which type of food to hide to protect their caches from being pilfered. To do so, jays had access to one caching location at a time (either in-view or out-of-view to the observing conspecific) and two types of food (one on which the observer was sated and one on which the observer was not sated). Following the general structure of Experiment 1, in the pre-feeding phase, the cacher bird was first able to see an observer eat one particular type of food (macadamia nuts or peanuts) to satiety. In the subsequent caching phase, the cacher bird was presented with a single caching tray and the two types of food (macadamia nuts and peanuts). To manipulate the observer's visual access to the caching location, the tray was placed behind an 'U'-shaped Perspex barrier (henceforth U-barrier) that consisted of two lateral panels and one central panel forming two angles of approximately 45° (see Materials and methods; *Figure 2*). The U-barrier was either transparent, thereby allowing the observer to see the caching location, or opaque, thereby preventing the observer from seeing the caching location.

Birds (n = 8; for details see Materials and methods) first received two familiarisation trials in private to ascertain that they were comfortable caching both types of food in a tray placed in proximity of each of the barriers. All birds except one reached the familiarisation criteria, namely to cache: (1) at

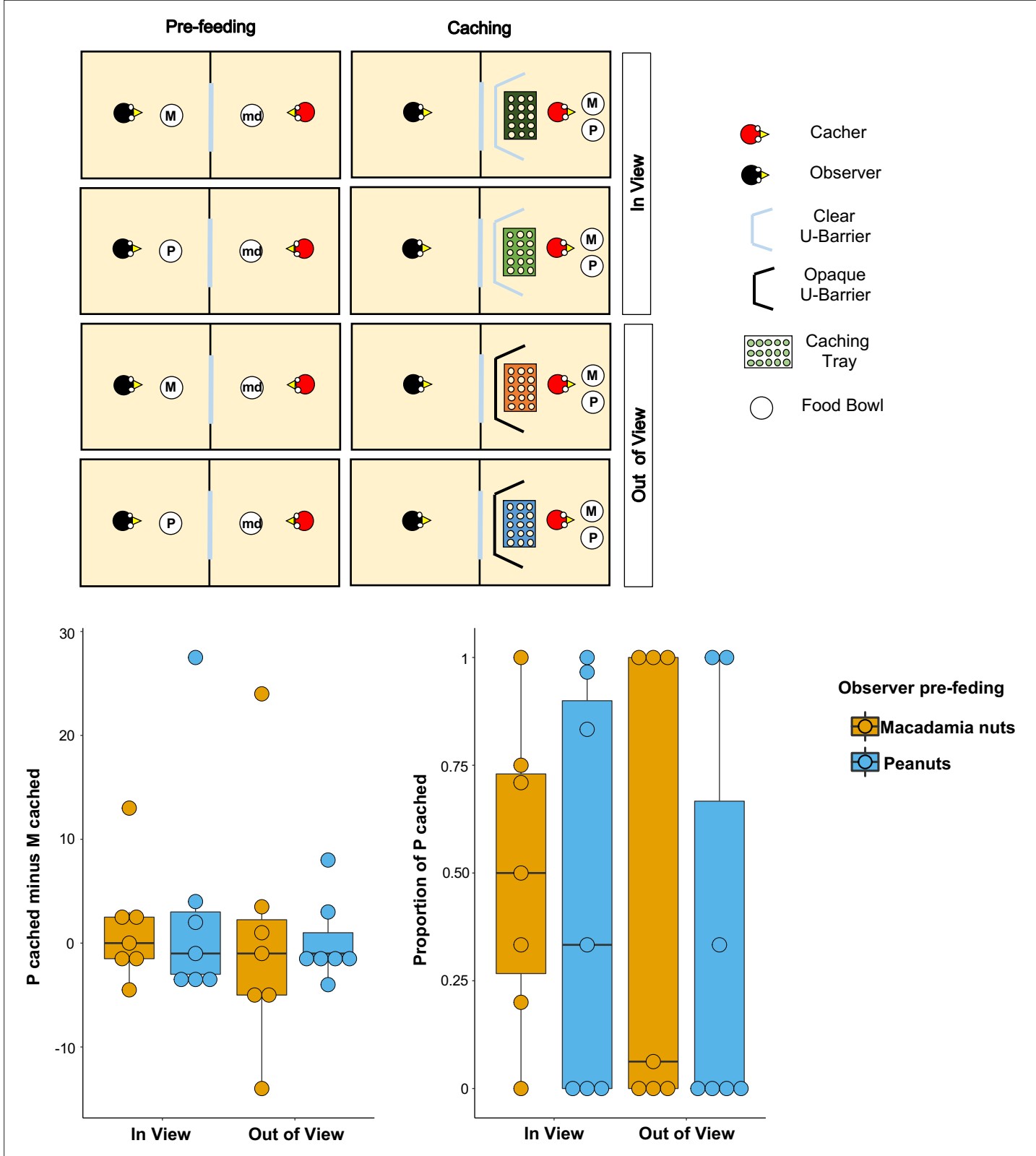

**Figure 2.** Methods and results of Experiment 2. Top panel: Top-view schematic representation of the set-up and procedure used in the test of Experiment 2. Trials were composed by a pre-feeding phase (left panels) and a caching phase (right panels). The cacher bird received two trials with the transparent U-barrier (*In-view* condition, top panels), and two trials with the opaque U-barrier (*Out-of-view* condition, bottom panels). Within each condition, trials differed in the type of food (either peanuts, P, or macadamia nuts, M) that was provided to the observer in the pre-feeding phase. The

*Figure 2 continued on next page*

*Figure 2 continued*

cacher bird was always presented with a handful of maintenance diet (md) in the pre-feeding phase of all trials. Bottom panel: Box and whisker plots of data in Experiment 2. The plot on the left shows the difference in the number of peanuts cached minus the number of macadamia nuts cached for each condition, whereas the plot on the right shows the proportion of P cached (out of total items cached) in each condition. In the *In-view* condition, the observer had visual access to the caching tray, whereas in the *Out-of-view* condition, the observer did not have visual access to the caching tray. The colour of the boxes in the plot differs on the basis of the type of food that was provided to the observer in the pre-feeding phase: blue denotes that the observer had been pre-fed P and orange denotes that the observer had been pre-fed M.

least one item of both types of food across the two trials, and (2) at least one item (of any type of food) in a tray placed in proximity of both the transparent and the opaque U-barrier. These birds (n = 7) were subsequently tested with the transparent barrier (*In-view* condition) and with the opaque barrier (*Out-of-view* condition). In each condition, the birds received two trials, one in which the observer was pre-fed one type of food and one in which it was pre-fed the other type (*Figure 2*).

If the jays can integrate information from the different cues available and which should correlate with the observer's desire and perspective, their caching pattern might be expected to meet two predictions. First, the jays' preference to cache P when the observer was sated on P relative to when the observer was sated on M, should be higher in the *In-view* than in the *Out-of-view* condition. This is because it is in the *In-view* condition that the observer can see the caching locations such that here the caching bird could protect its caches by caching preferentially more of the food that the observer is sated on. Second, in the *In-view* condition, the preference to cache P should be higher when the observer was sated on P than when the observer was sated on M. As in Experiment 1, both proportion and difference scores were used as indexes to analyse the birds' preference (for details see Analysis in the Materials and methods section).

Again, the results from the analyses using both indices were consistent. A comparison of the proportion of P cached between the *In-view* and the *Out-of-view* conditions – $[P_{cached}/(P_{cached} + M_{cached})]_{pre-fed P} - [P_{cached}/(P_{cached} + M_{cached})]_{pre-fed M}$ – did not detect a statistically significant difference ($Median_{In-view}$ = 0, $Median_{Out-of-view}$ = 0; Wilcoxon signed-rank test: n = 7, W = 3, p=0.83; *Figure 2*). In the *In-view* condition, no statistically significant difference in the proportion of P cached could be detected between the trials in which observer was sated on peanuts and the trials in which the observer was sated on macadamia nuts ($Median_{pre-fed P}$ = 0.33, $Median_{pre-fed M}$ = 0.5; Wilcoxon signed-rank test: n = 7, W = 3, p=0.83). Thus, neither prediction could be supported. An additional analysis of the same comparison for the *Out-of-view* condition also did not detect a statistically significant difference in the proportion of P cached between the trials ($Median_{pre-fed P}$ = 0, $Median_{pre-fed M}$ = 0.06; Wilcoxon signed-rank test: n = 7, W = 1, p=1).

The same pattern of results was found when the jays' preference was analysed using the other index, namely difference scores. No statistically significant difference in the preference to cache P over M when the observer is sated on P relatively to when the observer is sated on M – that is the difference of difference score: $[P_{cached} - M_{cached}]_{pre-fed P} - [P_{cached} - M_{cached}]_{pre-fed M}$ – was detected between the *In-view* and the *Out-of-view* conditions ($Median_{In-view}$ = –1, $Median_{Out-of-view}$ = 0; Wilcoxon signed-rank test: n = 7, W = –4, p=0.80; *Figure 2*). Furthermore, in the *In-view* condition, no statistically significant difference was detected between the trials in which observer was sated on peanuts and the trials in which the observer was sated on macadamia nuts ($Median_{pre-fed P}$ = –1, $Median_{pre-fed M}$ = 0; Wilcoxon signed-rank test: n = 7, W=-3, p=0.86). The additional analysis of the same comparison for the *Out-of-view* condition also did not detect a statistically significant difference between trials ($Median_{pre-fed P}$ = –1, $Median_{pre-fed M}$ = –1; Wilcoxon signed-rank test: n = 7, W=-1, p=1).

Consequently, as in Experiment 1, the observed data cannot be interpreted as support for the hypothesis that jays could integrate the information from multiple cues to protect their caches when these were most at risk of being pilfered by a conspecific. The two experiments thus yielded consistent results. However, a clear interpretation of the results is impeded by the likely low power of our designs to detect the smaller effect sizes that would be consistent with jays integrating both cues. This is likely due to the small sample size and limited number of trials per condition of our experiments, two features that – despite being relatively representative of the research in this area, including the previously published studies on this topic – may have produced imprecise estimates (*Farrar et al., 2020*; *Farrar and Ostojic, 2019*). Therefore, to strengthen our confidence that Eurasian jays may not be able to integrate multiple cues to protect their caches, it will be essential for future research to

conduct additional studies, ideally by employing larger sample sizes and procedures that can increase the precision of the analyses.

Although birds could have used multiple cues to guide their caching decisions in Experiments 1 and 2, they could also have adjusted their caching preference according to just one single type of cue, that is either the cues correlating with the observer's desire or the cues correlating with the observer's perspective. Both experiments used an experimental manipulation that, when applied separately, has already been reported – in previous studies – to have elicited a behavioural response that has been interpreted as a cache-protection strategy. Specifically, the caching phase of Experiment 1 involved the same procedure and set-up used by *Legg and Clayton, 2014*, except for the specific types and quantities of food provided to the jays. Similarly, the *In-view* condition of Experiment 2 and the *Seen* condition of *Ostojić et al., 2017*'s experiment employed the same procedure, with the exception that in the former the observers could see the caching location through a transparent barrier, whereas in the latter no barrier was present. However, in contrast to these previous studies, the results obtained in our experiments did not show a directional caching pattern in the predicted direction in these situations. Again, this may be a result of the low statistical power of our design, or possibly from the greater demands associated with tracking and integrating multiple cues to inform decision-making. However, the inconsistencies with previous research could also be due to previously reported effects not being reliable enough to form the basis of follow-up studies. Therefore, we conducted three further experiments to explore the reliability of the effects reported by *Legg and Clayton, 2014* and *Ostojić et al., 2017*. In Experiments 3 and 4, we attempted a replication of *Legg and Clayton, 2014*'s findings. In Experiment 3, birds were tested in one trial per condition, mirroring Experiment 1, while in Experiment 4, we conducted a complete direct replication of the original study that encompassed two trials in the *Private* condition and two trials in each of the two *Observed* conditions. Finally, in Experiment 5, we tested whether the presence or absence of a transparent barrier – that is the minor difference in the set-up between Experiment 2 and *Ostojić et al., 2017*'s experiment – may have affected the Eurasian jays' response in this caching situation.

## Exploring the reliability of caching strategies based on either the perspective or the current desire of a competitor

In Experiment 3, we investigated whether jays use information about an observer's visual perspective to protect their caches in a simplified version of *Legg and Clayton, 2014*'s experiment, that is the jays received only one trial in each of the two testing conditions. This mirrors the procedure in Experiment 1, where the same set-up was used and only one trial per testing condition was conducted. Following the original study, we presented cacher jays with two caching trays and manipulated the observer's visual access to cache locations by using the T-barrier. However, while *Legg and Clayton, 2014* tested jays in three conditions – *Observed by Dominant*, *Observed by Subordinate* and *Private* – and gave them two trials in each condition, in Experiment 3, jays (n = 8) received only two trials: one with a conspecific present in the adjacent compartment (*Observed* condition) and one with no conspecific present (*Private* condition). Seven birds met the inclusion criterion (see Material and methods for details).

The two analyses with the different indices yielded consistent results. The proportion of the items cached in the *out-of-view* tray was not significantly higher in the *Observed* condition than in the *Private* condition (Median$_{Observed}$ = 0.71, Median$_{Private}$ = 0.54; Wilcoxon signed-rank test, n = 7, W = 2, p$_{one-tailed}$ = 0.59). The same pattern was found when the difference score, that is, the number of items cached in the *out-of-view* tray minus the number of items cached in the *in-view* tray [Caches$_{out-of-view}$ – Caches$_{in-view}$], was analysed. The difference score was not significantly higher in the *Observed* than in the *Private* condition (Median$_{Observed}$ = 1, Median$_{Private}$ = 0.5; Wilcoxon signed-rank test, n = 7, W = 11, p$_{one-tailed}$ = 0.84).

In a subsequent experiment, Experiment 4, we conducted a direct replication of *Legg and Clayton, 2014*'s experiment. Here, the design and procedure were identical to those of the original study (see Materials and methods; *Figure 3*). Thus, in contrast to Experiment 3, in Experiment 4, jays were tested in three conditions – *Observed by Dominant*, *Observed by Subordinate* and *Private* – and received two trials in each condition. For this experiment only, we also tested the same colony of jays that originally participated in *Legg and Clayton, 2014*'s experiments. Because these birds had not recently participated in testing using the experimental set-up employed here and the T-barrier, we first conducted a

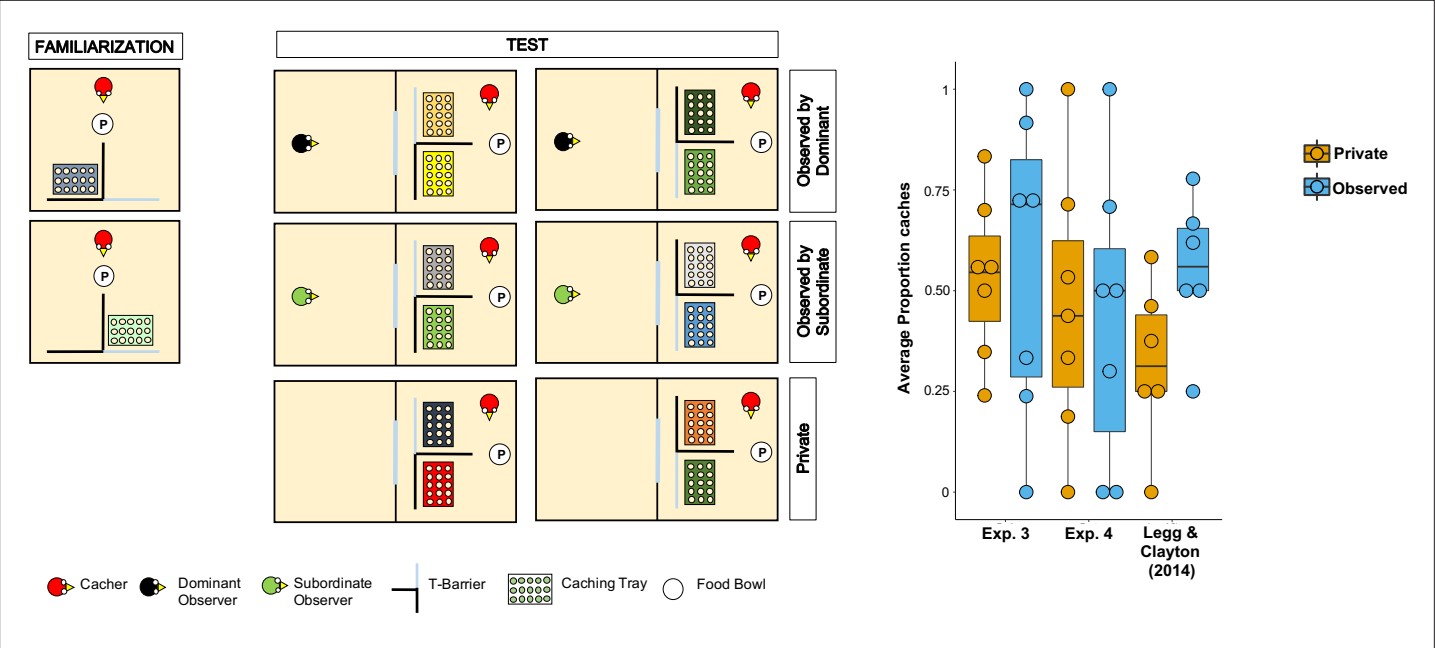

**Figure 3.** Methods and results of Experiment 4. Left: Top-view schematic representation of the set-up and procedure used in Experiment 4. In the familiarisation (left panels), the cacher bird received two trials, one in which the caching tray was placed near the opaque arm of the barrier (top panel), and one in which the tray was placed near the clear arm of the barrier (bottom panel). In the test (central and right panel), birds were tested in three conditions: *Observed by dominant*, *Observed by subordinate*, *Private*. In each condition, the cacher received two trials that differed in the orientation of the T-barrier. Right: Box and whisker plot of data in Experiments 3, 4, and in *Legg and Clayton, 2014*. The plot shows the average proportion of items cached in the *out-of-view* tray out of the total number of items cached, in the *Private* and *Observed* conditions. Note that data in the *Observed by dominant* and *Observed by subordinate* conditions were averaged for Experiment 4 and for *Legg and Clayton, 2014*'s study.

familiarisation that followed the same procedure as that used in Experiment 1. Nine birds passed the familiarisation and proceeded to the test. In this experiment only, we conducted the same analyses as for all other experiments (i.e., Wilcoxon signed-rank tests) but also an additional one, namely the same analysis (permutation tests for paired data) that was also used in *Legg and Clayton, 2014*. Again, a strong claim of an effect would require consistent results regardless of the analyses used.

In line with the original study, we found that the average number of total items cached across both trays was not significantly higher when the jays were observed by a conspecific than when they were in private (Permutation test, n = 9, Z = 0.79, p=0.43). Two birds cached no items in any of the *Private* and *Observed* trials, thereby they were excluded from further analyses of proportion scores because, given their performance, it was not possible to compare the proportion of items cached in the *out-of-view* tray between conditions. In the same analysis that was used by *Legg and Clayton, 2014*, the average proportion of items cached in the *out-of-view* tray was not significantly higher in the *Observed* condition than in the *Private* condition (Median$_{Observed}$ = 0.5, Median$_{Private}$ = 0.44; Permutation test, n = 7, Z = 0.15, p$_{one-tailed}$=0.56). The same results were found in the two analyses that used the same statistical test as in the other experiment reported in this study: average proportion of items cached in the *out-of-view* tray (Wilcoxon signed-rank test, n = 7, W = 2, p$_{one-tailed}$=0.59); average difference of the number of items cached in the *out-of-view* tray minus the number of items cached in the *in-view* tray (Median$_{Observed}$ = 0, Median$_{Private}$ = –0.5; Wilcoxon signed-rank test, n = 9, W = 13, p$_{one-tailed}$=0.20).

Taken together, Experiments 3 and 4 consistently did not detect the effect originally reported by *Legg and Clayton, 2014*, whereby Eurasian jays adjusted their caching pattern to the transparency and opaqueness of the barrier around the caching tray specifically when an observer was present during the caching event (*Figure 3*). In addition, the results from Experiments 3 and 4 appear consistent with the negative results from the *Different Food* condition in Experiment 1.

In Experiment 5, we investigated whether a minor difference in the set-up, that is the presence of a transparent barrier, may have caused the inconsistency in the results between Experiment 2 and

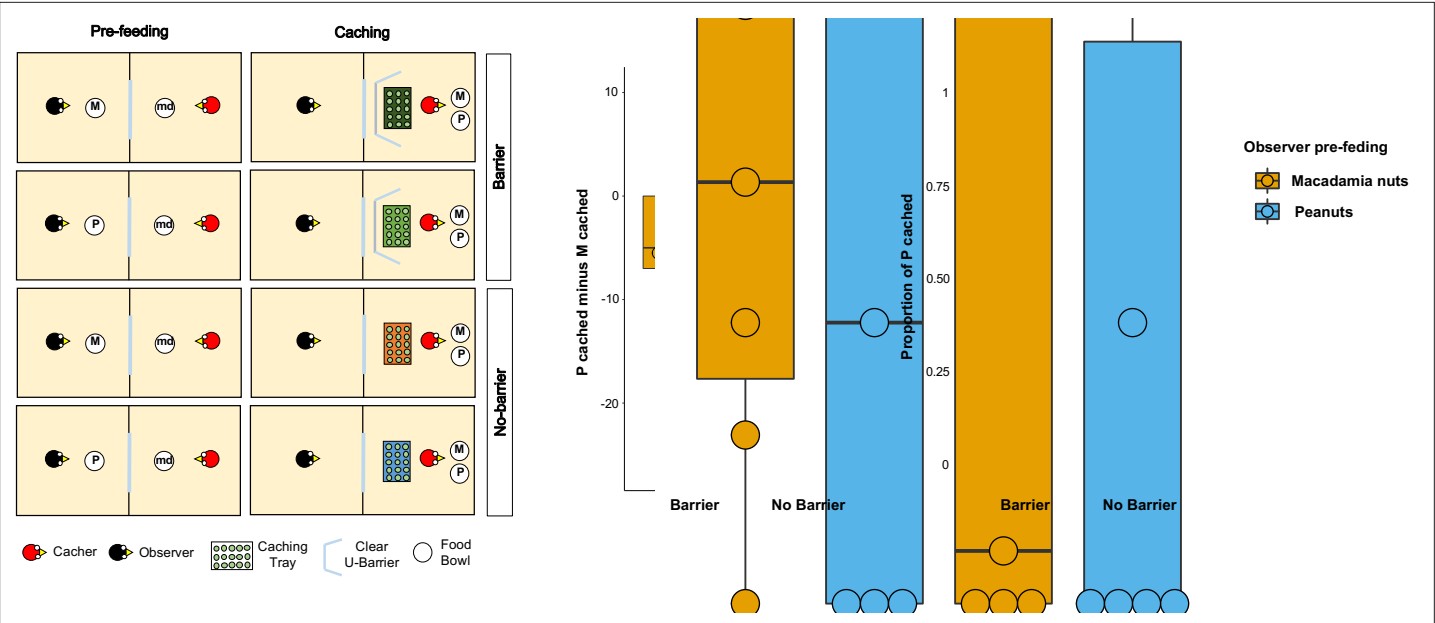

**Figure 4.** Methods and results of Experiments 5. Left: Top-view schematic representation of the set-up and procedure used in the test of Experiment 5. Trials were composed by a pre-feeding phase (left panels) and a caching phase (right panels). The cacher bird received two trials with the transparent U-barrier (*Barrier* condition, top panels), and two trials with no barrier (*No-barrier* condition, bottom panels). Within each condition, trials differed in the type of food (either peanuts, P, or macadamia nuts, M) that was provided to the observer in the pre-feeding phase. The cacher bird was always presented with a handful of maintenance diet (md) in the pre-feeding phase of all trials. Right: Box and whisker plots of data in Experiment 5. The central plot shows the difference in the number of peanuts cached minus the number of macadamia nuts cached in the *Barrier* condition (left) and *No-barrier* condition (right). The plot on the right shows the proportion of P cached (over the total number of items cached) in the two conditions. The colour of the boxes in the plot differs on the basis of the type of food that was provided to the observer in the pre-feeding phase: blue denotes that the observer had been pre-fed P and orange denotes that the observer had been pre-fed M.

the results reported in *Ostojić et al., 2017*'s study. To this end, we employed the same experimental set-up and procedures used in Experiment 2, except that here, in one condition, jays were presented with the transparent U-barrier (*Barrier* condition) and in another condition, with no barrier (*No-barrier* condition). All birds (n = 8) passed the familiarisation. In the test, one bird consistently cached no items, such that data of seven birds were analysed (see Materials and methods for details).

The two analyses using the two different indices yielded consistent results (*Figure 4*). No statistically significant difference could be detected in difference of the proportion of P cached when the observer was sated on P minus the proportion of P cached when observer was sated on M – [$P_{cached}$ / ($P_{cached}$+ $M_{cached}$)$_{pre-fed\ P}$] – [$P_{cached}$ /($P_{cached}$+ $M_{cached}$)$_{pre-fed\ M}$] – between the *Barrier* and *No-barrier* conditions (Median$_{Barrier}$ = 0, Median$_{No\ Barrier}$ = −0.04; Wilcoxon signed-rank test: n = 7, W = 11, p=0.18). In addition, in both conditions, no statistically significant difference could be detected in the proportion of P cached between the two pre-feeding trials (*Barrier* condition: Median$_{pre-fed\ P}$ = 0.17, Median$_{pre-fed\ M}$ = 0.1, Wilcoxon signedrank test, n = 7, W=-2, p$_{one-tailed}$=0.43; *No-barrier* condition: Median$_{pre-fed\ P}$ = 0,12, Median$_{pre-fed\ M}$ = 0,25, n = 7, W=-9, p$_{one-tailed}$=0.91).

The same patterns of results were observed when the difference score of the number of P cached minus the number of M cached was analysed. No statistically significant difference could be detected in the differences of difference score – [$P_{cached}$ − $M_{cached}$]$_{pre-fed\ P}$ – [$P_{cached}$ − $M_{cached}$]$_{pre-fed\ M}$ – between the *Barrier* and *No-barrier* conditions (Median$_{Barrier}$ = 0, Median$_{No\ Barrier}$ = 0; Wilcoxon signed-rank test: n = 7, W = 15, p=0.14). In addition, we detected no statistically significant in the difference score between the two pre-feeding trials, in either condition (*Barrier* condition: Median$_{pre-fed\ P}$ = −4, Median$_{pre-fed\ M}$ = −5, Wilcoxon signed-rank test, n = 7, W = 1, p$_{one-tailed}$ = 0.50; *No-barrier* condition: Median$_{pre-fed\ P}$ = −4, Median$_{pre-fed\ M}$ = −1, Wilcoxon signed-rank test n = 7, W=-9, p$_{one-tailed}$ = 0.91). Thus, the results from Experiment 5 cannot be interpreted as providing support for the idea that the presence of the barrier may be the reason why the results in Experiment 2 did not detect the same pattern as the one reported in *Ostojić et al., 2017*. Crucially, like in the *In-view* condition in Experiment 2, both

conditions in Experiment 5 also consistently could not detect the effect reported in *Ostojić et al., 2017*.

## Discussion

In Experiments 1 and 2, we investigated whether Eurasian jays can take into account two types of social cues simultaneously and perform the most advantageous behavioural output accordingly. Specifically, we tested whether caching birds can integrate information from cues correlating with a conspecific observer's desire and perspective to most effectively protect their caches. Consistently across these two experiments, we did not detect effects that would support such integration of information from different cues. In Experiment 1, jays did not show a higher preference for caching in the *out-of-view* tray when they could cache a food that was highly desired by the observer relative to when they could cache a food that was less desired by the observer. In addition, in the former case (i.e., *Different Food* condition), jays did not cache more in the *out-of-view* tray than expected by chance. In Experiment 2, jays did not show a higher preference for caching the food for which the observer had a decreased desire when the observer could see them relative to when the observer could not see them. In addition, in the former case (i.e., *In-view* condition), jays did not cache the food for which the observer had a decreased desire more than the food for which the observer had a higher desire.

The negative results we obtained in both experiments appear inconsistent with previous effects in the literature, despite the use of set-ups that were very similar to those used in the original studies. Specifically, the negative results in the *Different Food* condition in Experiment 1 appear incompatible with the effect reported in *Legg and Clayton, 2014*, where jays were found to preferentially cache in an *out-of-view* tray specifically when they were observed by a conspecific. Similarly, the negative results in the *In-view* condition in Experiment 2 appear incompatible with the effect reported in *Ostojić et al., 2017*, where jays were found to preferentially cache a specific food when the observer was pre-fed on that food relative to when the observer was pre-fed on a different food.

Thus, we conducted three follow-up experiments to explore the reliability of the two previous findings (*Legg and Clayton, 2014*; *Ostojić et al., 2017*) that our first two experiments were built on. In Experiments 3 and 4, we attempted to replicate the effect reported by *Legg and Clayton, 2014*, but – in contrast to the original study – no statistically significant difference between the experimental conditions was detected. Similarly, in Experiment 5 no statistically significant difference was detected between the experimental conditions, a result that contrasts with the effect reported by *Ostojić et al., 2017*. Thus, Experiments 3–5 also yielded negative results. However, evaluating the 'success' of a replication study from the statistical significance of a finding alone is overly simplistic, particularly for comparative cognition research, where – like in our experiment – the sample size of replication studies are often as equally small as that of the original studies (*Farrar et al., 2020*). Nevertheless, the finding that we could not detect any significant effect in line with the original experiments of *Legg and Clayton, 2014* and *Ostojić et al., 2017* across all five of our experiments was surprising, especially given that they were conducted in the same lab, with many of the same birds and experimenters. Specifically, in four of the seven tests of the hypothesis that the jays could use social cues to protect their caches, the results were not in the direction of the prediction: Experiment 1, prediction 1; Experiment 2, predictions 1 and 2; Experiment 5, prediction 2, *No-barrier* condition. In the remaining three tests in which we had a directional prediction – Experiments 3 and 4, and Experiment 5 prediction 2, *Barrier* condition – the effects were in the correct direction but were non-significant and much smaller than the previously reported effects.

It is not possible to provide a single, clear answer for why our study was unable to detect effects that are consistent with the previous literature. We propose two explanations that could have played a role, namely low power and our re-use of a unique sample of birds whose behaviour may change across time. First, each of our experiments likely had low power to detect theoretically meaningful, and perhaps the most theoretically plausible, effect sizes. This means that even if the jays were responding to the perspective and/or desires of the observer during our experiments, the effect size might have just been too small for us to detect. However, that we failed to find any significant results across five experiments with similar designs to the original studies shows that no extremely large effect sizes were present in our jays and suggests that the original effect sizes might be overestimated – which

**Table 1.** Individual data of the birds that participated in this study (Experiments 1–5).
The table reports also the individual data of the birds that participated in *Legg and Clayton, 2014*'s study and to the caching experiment by *Ostojić et al., 2017*.

| Colony | Bird | Sex | Born | Experiment |
|--------|------|-----|------|------------|
| 1 | Caracas | M | May 2006 | $1^{S, Ob}$, $2^{S, Ob}$, $3^{S, Ob}$, $4^{S, Ob}$, $5^{S}$, $O^{S, Ob}$ |
| 1 | Dublin | M | May 2006 | (1), $3^{Ob}$, $4^{S}$, $5^{S, Ob}$, $O^{S}$ |
| 1 | Jerusalem | F | May 2006 | $1^{S, Ob}$, $2^{S, Ob}$, $3^{S}$ |
| 1 | Lima | M | May 2006 | $1^{S}$, $2^{S}$, $3^{S, Ob}$, $4^{S}$, $5^{Ob, S}$ |
| 1 | Lisbon | M | May 2006 | $(1^{Ob})$, (2), $3^{Ob, S!}$, $4^{S, Ob}$, $5^{S!, Ob}$ |
| 1 | Quito | F | May 2006 | $1^{S}$, $2^{S}$, $3^{S}$, $4^{S, Ob}$, $5^{S}$ |
| 1 | Rome | F | May 2006 | $1^{S, Ob}$, $2^{Ob, S}$, $3^{S}$, $4^{S, Ob}$, $5^{Ob, S}$, $O^{S}$ |
| 1 | Washington | F | May 2006 | $1^{S}$, $2^{S, Ob}$, $3^{S, Ob}$, $4^{Ob, S}$, $5^{S}$, $O^{S}$ |
| 1 | Wellington | F | May 2006 | $1^{S, Ob}$, $2^{S, Ob}$, $3^{S}$, $4^{S, Ob}$, $5^{S, Ob}$, |
| 2 | Hunter | F | May 2008 | $4^{S}$, $L^{S}$ |
| 2 | Adlington | F | May 2008 | $(4^{Ob})$, $L^{Ob, S}$ |
| 2 | Webb | F | May 2008 | (4) |
| 2 | Hoy | M | May 2008 | (4), $L^{S, Ob}$, $O^{S, Ob}$ |
| 2 | Romero | M | May 2008 | (4), $O^{S}$ |
| 2 | Wilson | M | May 2008 | $L^{S, Ob}$ |
| 2 | Ohurougu | F | May 2008 | $L^{S, Ob}$ |
| 2 | Pendleton | M | May 2008 | $L^{S}$, $O^{S, Ob}$ |
| 2 | Ainslie | M | May 2008 | $L^{Ob}$ |
| 2 | Purchase | F | May 2008 | $L^{Ob}$ |

'L' refers to *Legg and Clayton, 2014*; 'O' refers to *Ostojić et al., 2017*. '()' means that the bird participated only in a preliminary phase of the experiment (i.e., familiarisation). This is relevant only to the experiments reported in this study but not to *Legg and Clayton, 2014*, *Ostojić et al., 2017*. 'Ob' means that the bird participated in the experiment as an observer. 'S' means that the bird participated in the experiment as a subject. '!' means that the bird did not complete the testing, such that its data was not included in the analyses. Note that the order in which 'Ob' and 'S' are reported describes the order in which the bird was used as observer and as subject (e.g., '$1^{Ob, S}$' means that the bird was used as observer before being used as subject in Experiment 1, whereas '$1^{S, Ob}$' means that the bird was used as observer after being used as subject in Experiment 1).

would be expected if the original studies came from a series of underpowered research combined with a publication bias (*Farrar et al., 2020*; *Fiedler and Prager, 2018*; *Hedges, 1984*).

Second, our five studies used the same populations of birds as tested in the previous studies (*Legg and Clayton, 2014*; *Ostojić et al., 2017*; *Table 1*), but they were around 5 years older. It is possible that the behaviour of these birds has changed over time, for example as a result of aging and/or learning effects. That is, the original studies might have reported effects without much overestimation, but when we tested the jays for these experiments, they may no longer have had the motivation to protect their caches. The overall number of cached items (as a proxy for cache motivation) does not appear too dissimilar in the current experiments and the previous studies (see *Figure 5* for a comparison of the number of items cached per bird per trial across all of the experiments). Thus, any differences relating to motivation of the birds would have to be specific to cache protection, rather than just caching.

With regard to the potential motivational changes due to ageing, it is possible that ontogenetic variations in sociality could influence the performance in caching tasks. Although data on corvids specifically are limited, research in primates has shown that motivations potentially relevant to cache protection, such as those driving socialisation and social influence, change dramatically throughout life, declining substantially in some taxa (*Machanda and Rosati, 2020*). A notable example of this

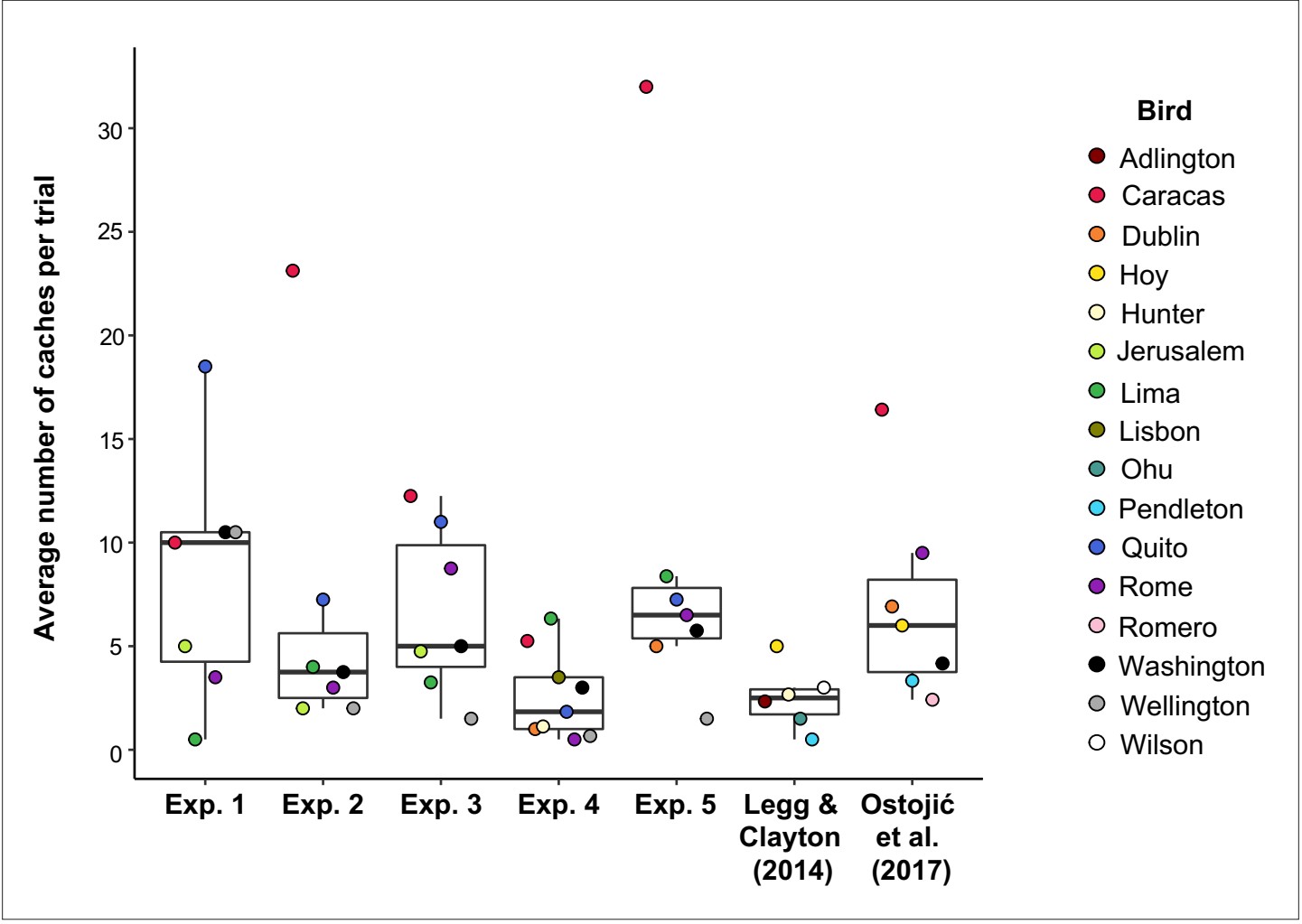

**Figure 5.** Box and whisker plot showing the mean number of items cached per trial in Experiments 1–5, *Legg and Clayton, 2014* and *Ostojić et al., 2017*. Note that *Ostojić et al., 2017* tested California scrub-jays and Eurasian jays but only Eurasian jays were relevant to the analysis, and therefore included in the plot. $Median_{Experiment\ 1} = 10$; $Median_{Experiment\ 2} = 3.75$; $Median_{Experiment\ 3} = 5$; $Median_{Experiment\ 4} = 1.83$; $Median_{Experiment\ 5} = 6.5$; $Median_{Legg\ \&\ Clayton\ (2014)} = 2.5$; $Median_{Ostojić\ et\ al.,\ 2017} = 6$.

process is provided by rhesus macaques, which, similarly to humans, show a reduced propensity to follow gaze in older age (*Rosati et al., 2016*). When it comes to corvids, the only data at the moment concern age-related shifts in non-social predisposition, such as neophobia (*Greggor et al., 2020*). Thus, age-related changes in the jays' motivation to attend to others' social cues are speculative, but in light of the negative results reported here, may present a line of enquiry for future studies.

With regard to potential changes due to prior experience and learning, previous studies with Western scrub-jays suggested that the caching behaviour itself is underpinned by a motivational system that acts as a compulsion to cache and that is not sensitive to reinforcement or other external conditions and that a second motivational system influences decision making during caching through sensitivity to conditions at cache retrieval (*Clayton et al., 2005*; *de Kort et al., 2007*). In our laboratory, caching experiments are conducted such that an observer is located in an adjacent compartment and cannot access the conspecific's caches, and in retrieval sessions the jays also receive feedback that their caches are unaffected by conditions at caching. If the jays' cognitive system tracks conditions at retrieval, then all experiments in which caches are unaffected, i.e., when jays regularly find them intact in the retrieval phase of a study, may favour a decrease in the birds' motivation to employ cache-protection strategies in the next similar situation. Thus, it cannot be excluded that the very first time when jays are presented with a novel set-up, their motivation to protect caches is higher than in subsequent presentations, but at the moment this must remain a speculation until studies directly test

this possibility. As such, already Experiments 1 and 2 but especially the replication attempts in Experiments 3–5 may have low validity, despite them using natural caching behaviours. This issue is likely exacerbated by the fact that our Eurasian jays regularly cache outside of testing time, such that laboratory conditions do not impair the natural intensity of caching behaviour exhibited by this species (*Goodwin, 1951*; *Goodwin, 1986*). In particular, our jays frequently save several food items (e.g., nuts, insect larvae) to take with them (e.g., store items in the crop) when released from the testing compartments to be cached in the aviary, as well as cache maintenance food (e.g., seeds, vegetables, eggs) during their normal daily activities given that their aviaries offer plenty of opportunities. This behaviour is in line with one of the experiments reported by *de Kort et al., 2007* in which the scrub-jays learned to inhibit caching at one location from which caches were pilfered if another location was available immediately afterwards in which caches were left intact.

To investigate the possible influences on the jays' motivational systems described above, one would ideally use a new sample of birds lacking an extensive experimental history with caching experiments. Unfortunately, it is unlikely that a study like the one above could be conducted in a near future on the same species of corvids because Eurasian jays are not currently housed in any other laboratory in the world, and there is no guarantee that a new colony will be acquired by our laboratory. There seem to be few other ways to test the motivation of our jays to display cache-protection behaviours because to validate their motivation to protect caches would require them to demonstrate the cache-protection behaviours that we set out to test. From negative results alone, it is unlikely that we can dissociate the possibility that the jays cannot display cache-protection behaviours from the possibility that the jays are not motivated to display them. Ultimately, we believe that these issues around the interplay between cognitive and motivational systems are not trivial and that they will need to be considered – alongside issues of low power – by researchers working in the field when addressing issues of reliability and validity of results, including replication attempts, especially given that many studies in the field of comparative cognition also rely on re-using the same samples of animals.

Our results conflict not just with the two studies that our research was built on *Legg and Clayton, 2014*; *Ostojić et al., 2017*, but also, more generally, with a larger body of literature on cache-protection strategies in corvids (*Bugnyar et al., 2016*; *Dally et al., 2004*; *Dally et al., 2005*; *Emery and Clayton, 2001*; *Heinrich and Pepper, 1998*) and Eurasian jays in particular (*Legg et al., 2016*; *Shaw and Clayton, 2013*). We were unable to elicit the cache-protection strategies that this literature implies are consistently observable across corvid species, including in our Eurasian jays. In addition to the two possible explanations raised above, it is important to discuss a third possibility. While it is possible our findings were only local failures to find these effects, it is also possible that the general research practices and methods that have produced the corvid social cognition literature are liable to producing unreliable findings or overestimated effect sizes. If the findings of the present study were shown to be indicative of a broader pattern in the field, it could be necessary to substantially revise our current understanding of corvid social cognition and their exhibition of particular cognitive phenomena. Crucially however, while our data clearly show a local failure to elicit previously reported effects, these data are not sufficient to draw strong conclusions beyond these local failures. This means that, despite posing a challenge to the reliability of the effects described by *Legg and Clayton, 2014* and *Ostojić et al., 2017*, in the absence of broader negative results, our findings cannot confute the notion that Eurasian jays are capable of employing cache-protection strategies by responding to cues about the visual perspective and current desire of a conspecific. Furthermore, in the lack of additional negative findings in other corvids or about different cognitive phenomena in the Eurasian jays, it is unclear to what extent our findings allow us to make inferences beyond the species and focus of this study. For instance, the fact that we could not find support that Eurasian jays respond to cues about others' current mental states in the caching context does not necessarily challenge the reliability of the evidence about similar behaviours in other contexts. For example, in the cooperative context of food sharing, the effect whereby males adjust the pattern of food shared with the female to her specific satiety (*Ostojić et al., 2013*) has been reliably shown by the same males in subsequent studies (*Legg, 2015*; *Ostojić et al., 2014*; *Ostojić et al., 2016*). One difference between the cache-protection and the food-sharing context may be the motivation of the jays to reliably exhibit the behaviour in question. Food sharing is an integral part of jays' courtship behaviour and is important not only in establishing but also maintaining the pair bond (*Goodwin, 1951*). Thus, males are likely to be motivated to respond to what the female likes or wants to eat across different time points. In

contrast, as discussed earlier, after jays' experience that their caches are not affected by any perceived risks, the motivation to employ cache-protection strategies in a very similar context may decrease.

We currently do not know how many other studies have produced negative cache-protection results but have not been published, and understanding the magnitude of the publication bias (*Fanelli, 2012*; *Scheel et al., 2020*) in this literature is therefore a necessary step to evaluating the evidential strength within the field. Concerning our failed replication of *Ostojić et al., 2017*, a slightly different reasoning applies as, to our knowledge, no similar study has been conducted in another laboratory. As such, we believe we have access to all the data on this topic. These are the current study, the *Ostojić et al., 2017* study, and a further, unpublished, replication attempt that also did not detect the originally reported effects (*Crosby, 2019*). Overall, the data on these effects seem too uncertain to draw any firm conclusions about Eurasian jay cognition.

Our difficulty with replicating previous research, even in the same laboratory as the original findings and with many of the same birds and experimenters, highlights two ways in which research on corvid social cognition – and likely, on comparative cognition research more broadly – could make progress. First, understanding the extent of publication bias in our literature is key to understanding their evidential value. Retrospectively, this may be achieved through meta-analysis techniques, and prospectively through effective pre-registering of hypothesis-testing research. Second, before building on findings, researchers can, where appropriate, include reliability tests into their research programmes. For example, replication and extension studies, conducted in a two-step process, such as have been traditionally employed in comparative cognition (*Beran, 2018*), present a useful procedure. First, the replication stage of the study tests whether an effect previously reported in the literature can be elicited under the circumstances (e.g., specific sample of individuals, testing facility or methodological procedure) in which the novel experimental question will be investigated. Only as a second step, the main study, the extension part, is performed. Such reliability tests may be especially important for previous findings where the effects of publication bias are unknown, as well as when the same animals are tested in follow-up tests (as is the case in this study) to probe the reliability of the behavioural patterns over time. To this end, particular efforts should be put into the designing of experiments, for instance by favouring within-subject designs as a way to account for individual-level variation, and by increasing the power of analyses through designs encompassing a larger number of test trials. We believe that these approaches are necessary to strengthen or reshape our understanding of corvid cognition. For instance, if subsequent evidence in Eurasian jays backs up the results of the current study, then belief that these corvids take into account information about the perspective or desire of a conspecific to protect their caches would diminish. However, because of the low likelihood of independent replication of many studies in comparative cognition, it is questionable whether it is possible to produce data that convincingly refute previous claims (*Boyle, 2021*). This situation in which some important and influential claims in the literature cannot be directly assessed presents a substantial challenge for comparative cognition research and warrants special and careful consideration and discussion within the field.

In conclusion, the current study presents five experiments that are inconsistent with the previous literature on caching in Eurasian jays. Across all experiments, the effects were non-significant, and often in the opposite direction to predictions derived from the published literature. This suggests that previous effect sizes are likely overestimated, or at the very least, that the effects cannot be consistently elicited in the same or similar samples of birds. Therefore, future studies should consider this new evidence when referring to the original studies for subsequent inferences about corvid social cognition. In Experiments 1 and 2, we investigated a follow-up question that assumed the reliability of previously reported statistical effects, which we later could not replicate in Experiments 3, 4, or 5. The current series of experiments demonstrate the necessity to investigate the uncertainty of such effects and to adjust the claims – including those in previously published literature – accordingly. In regard to the behavioural effects investigated in this study, the caching patterns interpreted as cache-protection strategies in *Legg and Clayton, 2014* and *Ostojić et al., 2017* do not seem to be reliable enough to form the basis for follow-up studies such as the ones reported in Experiments 1 and 2, at least in our sample of jays. It would be informative, but unfortunately not currently possible, to replicate these studies at other laboratories across the world.

# Materials and methods

## Subjects

Fourteen adult Eurasian jays from two separate colonies were tested in this study (*Table 1*). Most of the jays took part in multiple experiments and had previously been tested in experiments that involved caching in a similar set-up as that used in the current study (details about which jay participated in which experiment(s) are given in *Table 1*).

All of the jays were hand-raised, having been taken as chicks from wild nests or from the natural nests of birds in a breeding programme. The birds from each colony were housed as a group in large outdoor aviaries each measuring 20 m long × 10 m wide × 3 m high in Clayton's Comparative Cognition Lab at the Sub-Department of Animal Behaviour, University of Cambridge, Madingley, UK. At one end, the aviaries were divided such that birds had access to multiple smaller aviaries (approximately 6 × 2 × 3 m) and from these smaller aviaries birds could access indoor (colony 1) or fully sheltered (colony 2) testing compartments (2 × 1 × 3 m). Birds of colony 2 were housed in pairs in indoor cages until 2009 or 2010. Outside of testing the birds had ad libitum access to their maintenance diet of vegetables, eggs, seed, and fruits. Water was available at all times. All procedures were approved by the University of Cambridge Animal Ethics Committee (reference n. ZOO35/17).

## Experimental set-up

Birds were tested in the testing compartments measuring 2 m long × 1 m wide × 3 m high, which were accessible from the smaller aviaries through flap windows. In trials requiring the presence of an observer, two birds – a cacher bird and an observer bird – were located in adjacent compartments. These compartments were separated by wire mesh and additional opaque sheeting. A little mesh window (30 × 55 cm) was not covered by the opaque sheeting and through it the birds had visual access to the adjacent compartment. Testing compartments contained a suspended platform (1 × 1 m) approximately 1 m from the ground, onto which food bowls, caching trays, and Perspex barriers could be placed. Each type of food used in the experiments was presented in a bowl of a specific colour, and these colours were kept consistent for all birds to minimise the likelihood of experimenter errors. Rectangular seedling trays (5 × 3 pots filled with sand) were used as caching trays. Trays were painted different colours and were trial specific to minimise the probability that birds' caching behaviour in one trial would be influenced by its memory from previous trials.

In Experiments 1, 3, and 4, a T-barrier was used to manipulate the observer's visual access to the caching trays. It was the same T-barrier that *Legg and Clayton, 2014* used. This barrier consisted of three plastic panels (25 × 40 cm) forming two arms and one stem. One arm of the 'T' was constructed out of transparent Perspex, while the other arm and the stem were constructed out of white opaque Perspex. The T-barrier could be placed around two caching trays in the cacher's compartment, such that the observer could see the tray behind the transparent arm (*in-view* tray) but could not see the tray behind the opaque arm (*out-of-view* tray). Due to the height of the barrier, the observer could always see the cacher when the latter was standing upright in proximity to the trays. However, the observer could not see the exact location where the cacher hid the food when it was caching in the *out-of-view* tray.

In Experiments 2 and 5, a U-barrier was used to manipulate the observer's visual access to the caching trays. The barrier consisted of two lateral Perspex panels (26 × 25 cm) and one central Perspex panel (53 × 25 cm) forming two angles of approximately 45°. In Experiment 2, we used two U-barriers, one made of transparent Perspex and another made of white opaque Perspex. In Experiment 5, only the transparent barrier was used. The U-barrier was placed around a single tray in the cacher's compartment, and if opaque, it impaired the observer's visual access to the caching tray.

## General procedures

In all experiments, the birds' maintenance diet was removed from the aviary approximately 1.5 hr prior to the start of each trial to ensure that the birds were mildly hungry and thus likely to interact with food provided during testing.

## Familiarisation

In all experiments in which the birds had not experienced the set-up and apparatuses just prior to testing (i.e., in Experiments 1, 2, 4, and 5), a familiarisation procedure was conducted to ascertain that birds were comfortable caching in trays placed in proximity of the respective barriers (see Specific Procedures for further details). During the familiarisation, each bird was tested in isolation, i.e., with no other birds present in the test area. Compartments used during the familiarisation were not used in the test phase to minimise the probability of carry-over effects.

## Test

In Experiments 1, 2, and 5, test trials involved a pre-feeding phase followed by a caching phase. Before the start of a test trial, two birds (a cacher and an observer) were given access to two adjacent compartments. Subsequently, the experimenter placed a bowl containing the pre-feeding food (macadamia nuts or peanuts) on the suspended platform in the observer's compartment and a bowl containing a handful of maintenance diet on the platform in the cacher's compartment (*Figures 1, 2 and 4*). Both bowls were placed in front of the mesh window to ensure that the birds could see each other whilst eating and to maximise the likelihood that the cacher could see on which food the observer was pre-fed. The experimenter then left the test room and the birds could eat the pre-feeding food for 15 min. Next, the experimenter entered the test room again and removed the bowls as well as any food remains on the platforms. In the subsequent caching phase, the caching trays, as well as the barrier and the food bowl, were positioned in front of the mesh window in the cacher's compartment (*Figures 1, 2 and 4*). The experimenter then left the test room and the birds were given 15 minutes during which the cacher could eat and cache the food in the trays. In Experiments 3 and 4, the test trials involved only a caching phase (*Figure 3*). Before the start of a trial, the cacher bird was given access to the testing compartment where the caching trays and the T-barrier had already been positioned. In the test trials of the *Observed* condition, a second bird (i.e., the observer) was also induced to enter the adjacent compartment. Subsequently, a food bowl was placed on the suspended platform in the cacher's compartment. The experimenter then left the test room and the cacher was given the opportunity to eat and cache food for 15 min.

At the end of each familiarisation and test trial, the experimenter opened the flap windows to allow the bird(s) to re-join the rest of the group in the aviary and recorded the amount of food eaten and the number and location of caches by manually checking the food bowls and trays. Approximately 3 hr after each trial, the cacher was allowed to re-enter the caching compartment. No other birds were present in the test area and the flap window was kept open so that the bird had access not only to the test compartment but also to the adjacent smaller aviary. Note that the door connecting the small aviary to the main aviary was kept closed such that no other bird could enter the cacher's aviary or compartment. The cacher could retrieve the hidden items and re-cache them in the compartment and in the adjacent small aviary. This retrieval phase was conducted only to reduce the probability that birds would stop caching in the trays, and thus these data were not analysed. Birds received a single test trial per day.

## Specific procedures

### Experiment 1

#### Familiarisation

Birds (n = 9; *Table 1*) received two familiarisation trials on two separate days to ensure that they were comfortable caching in trays when these were placed in proximity of each of the two arms of the T-barrier. On each trial, the bird was presented with the T-barrier, a single caching tray and a food bowl containing either 50 macadamia nut halves (M) or 50 whole peanuts with skin (P). The type of food (macadamia nuts or peanuts) was randomly assigned to birds but each bird was provided with the same type of food in both trials. The bird was given the opportunity to eat and cache for 15 min. On one trial, the tray was placed behind the opaque arm of the T-barrier, and on the other trial, it was placed behind the transparent arm. The order in which birds experienced the tray in the two locations was counterbalanced across birds. The orientation of the barrier within the compartment was different from that later used during testing and was kept consistent for each bird across the two familiarisation trials (*Figure 1*). This procedure was chosen to ensure that the birds were not more familiar with one of the two orientations of the barrier in a specific spatial set-up (e.g., opaque arm facing the outdoor

aviary) in the subsequent test. To proceed to the test, birds had to cache at least one food item in the tray on each trial. If a bird did not meet this criterion, it was excluded from further testing. All birds except two (i.e., Dublin and Lisbon; *Table 1*) passed the familiarisation and proceeded to the test.

## Test

During the pre-feeding phase, cachers (n = 7; *Table 1*) could see a conspecific eat a specific type of food: either the same type of food they were going to receive in the subsequent caching phase (*Same Food* condition) or a different one (*Different Food* condition; *Figure 1*). The order in which the birds experienced the *Different Food* and *Same Food* conditions was counterbalanced across birds. In the subsequent caching phase, cachers were provided with the same food used in the familiarisation and with two caching trays, each one placed behind one of the two arms of the T-barrier (*Figure 1*). The food given to the observer during the pre-feeding phase and to the cacher during the caching phase was either 50 macadamia nut halves or 50 whole peanuts with skin. All birds received one trial per condition, that is, two test trials in total. If a bird cached no items on both trials, it was paired with a different observer and the two trials were repeated. If it again did not cache on both trials, these data were not included in the analysis. In contrast, if the bird cached with the second observer, then these data were included in the analysis. This procedure was decided during data collection, after one bird (Lima) did not cache any food across both trials, but before the analysis was conducted. For all other birds, test trials were not repeated. The analysis included the data of all seven birds. Experiment 1 was conducted from October to November 2017 by LO, BF, and PA.

## Experiment 2

### Familiarisation

Birds (n = 8; *Table 1*) received two familiarisation trials on two separate days to ascertain that they were comfortable caching both types of food (macadamia nuts and peanuts) in a tray placed in proximity of each of the U-barriers (transparent and opaque). On each trial, the bird was presented with a U-barrier, a single caching tray and two food bowls, which were presented sequentially. The food bowls contained either 50 macadamia nut halves or 50 whole peanuts with skin. The bird was given the opportunity to eat and cache for 20 min: during the first 10 minutes it was provided with one type of food and during the next 10 min with the other. The order in which the birds experienced the two types of foods was counterbalanced across birds and across trials, such that each bird experienced one order on their first trial and the opposite order on their second trial. On each trial, the barrier was either transparent or opaque. The order in which the birds experienced the two types of the U-barrier was counterbalanced across birds. To proceed to testing, birds had to (1) cache at least one item on each trial and (2) cache at least one item of each kind of food across the two trials. If a bird did not meet these criteria, it was excluded from further testing. All birds except one (Lisbon; *Table 1*) passed the familiarisation and proceeded to the test.

### Test

The pre-feeding phase was conducted in the same way as in Experiment 1. In the subsequent caching phase, the cacher was provided with a single tray placed within the U-barrier and two food bowls (one containing 50 macadamia nut halves and the other containing 50 whole peanuts with skin). Birds (n = 7; *Table 1*) received four trials in total: there were two conditions, namely the *In-view* condition (transparent U-barrier) and the *Out-of-view* condition (opaque U-barrier), and within each condition, there were two trials, one in which the observer was pre-fed on one type of food (e.g., macadamia nuts) and one in which the observer was pre-fed on the other type food (e.g., peanuts). Birds first received both trials of one condition, and then the two trials of the other condition. The order of conditions was counterbalanced across birds. The order in which the observer was pre-fed on the two types of food was counterbalanced across birds within condition, but kept consistent across conditions (i.e., the order of the two trials was the same in both conditions). If a bird did not cache any items in one or two trials, those trials were repeated at the end of the experiment. However, if a bird did not cache any items in more than two trials, that bird was not tested any further and was excluded from the analysis. A specific trial was repeated for a maximum of two times, such that a bird was excluded from the analysis if it cached no item in both repeated trials. It was necessary to repeat trials for three birds (i.e., Jerusalem: two repeated trials; Lima: two repeated trials; Rome: one repeated trial). The

analysis included the data of all seven birds. Experiment 2 was conducted from January to February 2018 by PA.

## Experiment 3
### Familiarisation
Here, we conducted no familiarisation because this experiment took place shortly after Experiment 1 (which also involved the T-barrier) and involved the same birds. Thus, participation in Experiment 1 already insured that birds were comfortable caching in trays next to the T-barrier.

### Test
The procedure of the test was simplified from the procedure in *Legg and Clayton, 2014*. The main difference was that there was only one trial per condition. In addition, we used either 50 whole peanuts with skin or macadamia nut halves (counterbalanced across birds) as food for the cachers, whereas the original study used 30 peanut halves. Birds (n = 8; *Table 1*) were given two trials in total: one with an observer present in the adjacent compartment (*Observed* condition) and one in which no observer was present (*Private* condition). The order of the conditions was counterbalanced across birds. On each trial, two trays were positioned behind the T-barrier, one behind the opaque and one behind the transparent arm. The orientation of the T-barrier was counterbalanced across birds but kept constant across trials for each bird. The cacher was given 15 min during which it could each or cache the food. If a bird did not cache on a trial, the trial was repeated. If the bird did not cache again, the data were not included in the analysis. A trial was repeated for one bird (Caracas). Due to timing constraints, another bird (Lisbon) was not given the possibility to repeat the trials in which no item was cached. Thus, although this was not pre-specified as an exclusion criterion, Lisbon's data were excluded from the analysis before it was conducted. The analysis included the data of seven birds, that is, all birds except one (Lisbon). Experiment 3 was conducted in December 2017 by LO.

## Experiment 4
### Dominance hierarchy
In *Legg and Clayton, 2014*'s experiment, cacher birds received four trials in the *Observed* condition: two trials in which they were observed by a higher ranked individual (*Observed by dominant* condition), and two trials in which they were observed by a lower ranked individual (*Observed by subordinate* condition). Thus, to replicate the original design, it was necessary to determine the dominance hierarchy within each colony. To this end, ad libitum observations were conducted for each colony. Birds were observed as a group in the main outdoor aviaries after their maintenance diet had been removed for approximately 2 hr. For each observation session, maintenance diet was presented on a single food platform in the aviary. This procedure was chosen to increase activity within the colony. To solicit competitive interactions among birds, higher value food items (e.g., wax worm larvae, *Galleria mellonella*) were also presented in a bowl or scattered around on the floor of the aviary. The identity of both actor and recipient involved in any displacement (i.e., Bird X approaches Bird Y causing Bird Y to leave) was recorded. If necessary, to obtain data for all birds, higher ranked birds were locked into separate compartments to favour interactions among lower ranked birds. Observation sessions were conducted on multiple days, until data were collected to establish a clear social hierarchy within each colony. Sessions lasted approximately 40 min each. Dominance hierarchy data were not collected for colony 2 because only one bird of this group (Hunter) passed the familiarisation.

### Familiarisation
Unlike birds from colony 1, birds of colony 2 had not recently had any experience with the T-barrier. Thus, all birds (n = 13; *Table 1*) received two familiarisation trials on separate days to ascertain that the birds were comfortable caching in proximity of both the transparent and the opaque arm of the T-barrier (see also pre-registration addition from 8 Dec 2018). This familiarisation followed the same procedure as the familiarisation in Experiment 1, except that here, each bird was provided with 30 peanut halves as in the original study *Legg and Clayton, 2014*. To proceed to the test, birds were required to cache at least one food item in each of the two familiarisation trials. If no item was cached in a trial, then that trial was repeated for a maximum of two times. Thus, a bird could receive a maximum of six trials in total. The repeated trials were conducted at the end (e.g., if a bird's first trial

had to be repeated, then the bird received the second, pre-planned trial on day 2, and subsequently it received the first trial again on day 3). It was necessary to repeat trials for two birds (i.e., Lisbon: one repeated trial; Lima: one repeated trial). Nine birds (*Table 1*) passed the familiarisation and proceeded to the test. Due to an experimenter's error, the raw data for one bird (Hunter) in the familiarisation was not archived.

## Test

Following the procedure in *Legg and Clayton, 2014*, birds (n = 9) received six trials in total. There were three conditions (*Private* condition, *Observed by dominant* condition, and *Observed by subordinate* condition) and in each condition, the cacher experienced the T-barrier in two different orientations (opaque arm of the barrier was facing the outdoor aviary, transparent arm facing outdoor aviary) on two separate trials. Each bird was first tested in all conditions with the barrier being kept consistent in one specific orientation, then subsequently received the remaining trials with the barrier being kept consistent in the alternative orientation. The order in which the two orientations of the barrier were experienced was counterbalanced among birds.

On each trial, the cacher was given access to the testing compartment and presented with the T-barrier, the *in-view* tray (i.e., the tray placed behind the transparent arm of the barrier), the *out-of-view* tray (i.e., the tray placed behind the opaque arm of the barrier), and a bowl containing 30 peanuts halves. The bowl was placed close to the stem of the 'T' such that it was equidistant from the two caching trays (*Figure 4*). The cacher could eat and cache for 15 min and was subsequently released back into the aviary.

All birds were tested in all three conditions, except the highest and lowest ranked bird in each colony. The former could only be tested in the *Private* and *Observed by subordinate* conditions and the latter could only be tested in the *Private* and *Observed by dominant* conditions. Thus, most birds received six trials in total (two trials per condition), whereas the highest and lowest ranked individuals in each colony received four trials in total because they could only be tested in two of the three conditions. In line with the procedure of the original study, test trials were not repeated if the bird cached no item. However, we decided to repeat a specific trial (Dublin's first trial in the *Observed by dominant* condition) because the bird that served as observed (Rome) appeared to experience issues with flying. This trial was repeated with a different observer after the remaining pre-planned trials were completed. The analysis included the data of all nine birds. This experiment was pre-registered on the Open Science Framework (https://osf.io/8p4tx/). The pre-registration was conducted after the familiarisation was completed but before the start of the test. Experiment 4 was conducted from October to December 2018 by PA (colony 1) and Rachel Crosby (colony 2).

## Experiment 5

### Familiarisation

Birds (n = 8; *Table 1*) received two familiarisation trials on separate days to ensure that they were motivated to cache both types of food and were comfortable caching in a tray both when it was positioned close to the U-barrier and when no barrier was present. Thus, the familiarisation followed the procedure of the familiarisation in Experiment 2, except that here, one trial involved the transparent U-barrier and the other one no barrier. To proceed to testing, birds had to (1) cache at least one item on each trial (i.e., both with barrier present and with no barrier present), and (2) cache at least one item of each type of food across the two trials. If no item was cached in a trial, then that trial was repeated for a maximum of two times. Thus, a bird could receive a maximum of six trials in total. The repeated trials were conducted at the end: for example, if a bird's first trial had to be repeated, then this bird received the second, pre-planned trial on day 2, and subsequently it received the first trial again on day 3. It was necessary to repeat trials for two birds (i.e., Lisbon: one repeated trial; Wellington: one repeated trial). All eight birds (*Table 1*) passed the familiarisation and proceeded to the test.

### Test

The procedure in the test phase was the same as in Experiment 2, except that instead of two different U-barriers being used (transparent and opaque), here there was either a clear U-barrier (*Barrier* condition) or no barrier at all (*No-barrier* condition; *Figure 4*). Birds first received both trials (observer pre-fed macadamia nuts and observer pre-fed peanuts) of one condition (e.g., *Barrier* condition),

and then the two trials of the other condition. The order in which the two conditions were conducted was counterbalanced across birds. The order in which observers were pre-fed the two kinds of food within a condition was counterbalanced across birds, but kept consistent across conditions such that the order of the two trials for each bird was the same in both conditions. If a bird did not cache any items in one or two trials, those trials were repeated at the end of the experiment. Each trial could be repeated no more than two times (i.e., three attempts in total). If a bird cached no item in more than two trials, that bird was not tested any further and was excluded from the analysis. It was necessary to repeat trials for two birds (i.e., Quito: one trial; Wellington: two trials). The analysis included the data of seven birds (*Table 1*), that is all birds except one (Lisbon), that cached no items in three trials. This experiment was pre-registered on the Open Science Framework (https://osf.io/8p4tx/). Experiment 5 was conducted in November 2018 by PA.

## Data collection

In all experiments, we recorded the number and type of food items cached on each trial by manually checking the trays. The experimenters were not blind to the conditions while counting the food items. These data were used to test whether the birds had a preference for caching a specific type of food or for caching in a specific tray. In all experiments, we also recorded (1) the number of items taken from the bowl by observers (during pre-feeding) and by cachers and (2) the number of items recovered by cachers during retrieval sessions. These data were collected such that all data available for each trial are archived and available, but these data were not relevant to the experimental question so that they were not analysed.

## Statistical analysis

The birds' preference for a specific type of food or tray was analysed according to two indices: proportion scores (e.g., the proportion of items cached in one location out of total number of items cached in both locations) and difference scores (e.g., number of items cached in one location minus the number of items cached in the other location). As stated in the pre-registrations of Experiments 4 and 5 (https://osf.io/8p4tx/), we originally planned to analyse the data of all five experiments only through proportion scores. However, when a bird caches no item in a trial, then the individual performance in that specific trial cannot be analysed through the proportion scores, yet it can still be analysed through the difference scores. This issue is relevant only to Experiment 4, where – in line with the procedure of the original study by *Legg and Clayton, 2014*, and in contrast with the procedure of Experiments 1, 2, 3, and 5 – the trials in which no item was cached were not repeated. Nevertheless, after the study was conducted, we decided to analyse the data of all experiments – not only the data of Experiment 4 – also by using the difference scores. We reasoned that, if there are large discrepancies between the results obtained with both types of indices, then this may be important information regarding the robustness of any effects because such discrepancies would show that results from small sample sizes are easily susceptible to change based on the type of analysis used.

No power analysis was conducted to establish an appropriate sample size for the study. In each experiment, we used all the individuals out of all the Eurasian jays housed in the facility, that were available for testing at the time when the experiment was being conducted and were familiar with the general set-up of caching experiments (i.e., the jays that would cache in caching trays placed in indoor testing compartments).

## Experiment 1

For each trial, we calculated the proportion of items cached in the *out-of-view* tray out of the total number of items cached in the *out-of-view* and *in-view* trays [Caches$_{out-of-view}$/(Caches$_{out-of-view}$+ Caches$_{in}$)]. In parallel, for each trial we calculated the difference score, that is, the number of items cached in the *out-of-view* tray minus the number of items cached in the *in-view* tray [Caches$_{out-of-view}$ − Caches$_{in-view}$]. Both indices indicate a preference for caching in the *out-view* tray over the *in-view* tray. Wilcoxon signed-rank tests were used to test whether the two indices of preference for caching in the *out-of-view* tray differed between the *Same Food* condition and the *Different Food* condition. Further, in the *Different Food* condition, one-sample Wilcoxon signed-rank tests were used to test whether the preference for caching in the *out-of-view* tray was different from that expected by chance, that is, 0.5 for the proportion score, and 0 for the difference score. As an additional, exploratory analysis,

one-sample Wilcoxon signed-rank tests were used to investigate whether the preference for caching in the *out-of-view* tray differed from chance (again, 0.5 for the proportion score and 0 for the difference score) in *Same Food* condition.

### Experiment 2

For each trial, we calculated (1) the proportion of peanuts (P) cached out of the total number of peanuts and macadamia nuts (M) cached [$P_{cached}/(P_{cached} + M_{cached})$] and (2) the difference score, i.e., the number of P cached minus the number of M cached [$P_{cached} - M_{cached}$]. These scores indicate a potential preference for caching P over M. For each condition (*In-view* and *Out-of-view* condition), we further calculated (1) the difference of proportions score, namely the proportion of P cached when the observer was pre-fed on P minus the proportion of P cached when the observer was pre-fed on M: [$P_{cached}/(P_{cached} + M_{cached})]_{pre-fed\ P}$ − [$P_{cached}/(P_{cached} + M_{cached})]_{pre-fed\ M}$ and (2) the difference of difference score, namely the difference score when the observer was pre-fed on P minus the difference score when the observer was pre-fed on M: [$P_{cached} - M_{cached}]_{pre-fed\ P}$ − [$P_{cached} - M_{cached}]_{pre-fed\ M}$. These scores indicate a potential preference to cache P over M when the observer was sated on P relative to when the observer was sated on M. Consequently, Wilcoxon signed-rank tests were used to test whether the difference of proportion scores and the difference of differences scores were statistically different between the *In-view* and in the *Out-of-view* conditions. In addition, Wilcoxon signed-rank tests were also used to test whether – in the *In-view* condition – the proportion score and the difference score differed when the observer was pre-fed on P relative to when the observer was pre-fed on M. The same tests were also conducted in the *Out-of-view* condition as an exploratory analysis.

### Experiment 3

As in Experiment 1, for each trial we calculated the proportion of items cached in the *out-of-view* tray – [$Caches_{out-of-view}/(Caches_{out-of-view} + Caches_{in-view})$] – and the difference between the number of items cached in the *out-of-view* tray and the number of items cached in the *in-view* tray – [$Caches_{out-of-view} - Caches_{in-view}$]. Subsequently, we used one-tailed Wilcoxon signed-rank tests to investigate whether each of the two indices was significantly higher in the *Observed* condition than in the *Private* condition.

### Experiment 4

As in Experiment 1, for each trial we calculated the proportion of items cached in the *out-of-view* tray – [$Caches_{out-of-view}/(Caches_{out-of-view} + Caches_{in-view})$] – and the difference between the number of items cached in the *out-of-view* tray and the number of items cached in the *in-view* tray – [$Caches_{out-of-view} - Caches_{in-view}$]. Following *Legg and Clayton, 2014*, for both indexes we calculated individual mean values in the *Private* condition and *Observed* condition. In the latter case, all trials in which an observer was present (i.e., *Observed by dominant* condition and *Observed by subordinate* condition) were taken into account to calculate individual mean values. It should be noted that for the proportion score, the actual number of trials that could be used to calculate the individual means in each condition was not consistent for all birds. This is because the trials in which no item was cached across both trays had to be excluded. We also calculated the mean number of caches made in both trays by each individual, in each condition.

The data were analysed using the same statistical analysis used by *Legg and Clayton, 2014*. In particular, we used a one-tailed permutation test to investigate whether the individual mean of the proportion of the caches in the *out-of-view* tray was higher in the *Observed* condition than in the *Private* condition. Following *Legg and Clayton, 2014*, we also tested whether the average number of items cached across both trays differed between the *Observed* and *Private* conditions by using a two-tailed permutation test.

In parallel, we also analysed the data using Wilcoxon signed-rank tests like in all other experiments reported here. In particular, one-tailed tests were used to investigate whether individual mean values (for both the proportion score and the difference score) were significantly higher in the *Observed* condition than in the *Private* condition.

### Experiment 5

As in Experiment 2, for each trial we calculated the proportion of P cached – [$P_{cached}/(P_{cached} + M_{cached})$] – and the difference of P cached – [$P_{cached} - M_{cached}$]. Furthermore, as in Experiment 2, we also calculated

for each condition (*Barrier* and *No-barrier* conditions) the difference of proportion scores – $[P_{cached}/(P_{cached} + M_{cached})]_{pre-fed\ P}$ – $[P_{cached}/(P_{cached} + M_{cached})]_{pre-fed\ M}$ – and the difference of differences scores – $[P_{cached} - M_{cached}]_{pre-fed\ P}$ – $[P_{cached} - M_{cached}]_{pre-fed\ M}$. Consequently, we used Wilcoxon signed-rank tests to investigate whether the difference of proportions score and the difference of differences score were statistically different between the *Barrier* and *No-barrier* conditions. In addition, one-tailed Wilcoxon signed-rank tests were also used to test whether – in each condition – the proportion score and the difference score were higher when the observer was pre-fed P relative to when the observer was pre-fed M.

All statistical analyses were performed in R (R.3.5) using the RStudio 1.1.447 wrapper (***R Studio Team, 2018***). Permutation tests were conducted with the package *coin* (***Hothorn et al., 2006***). All tests were two tailed, unless stated otherwise. Alpha was set to 0.05.

## Acknowledgements

We are grateful to Rachel Crosby for collecting data for Experiment 4 in colony 2 and for her feedback on the manuscript. FUNDING During the preparation of the manuscript, PA received support from the Leverhulme Trust (Grant reference: SAS-2020–004\10). BGF was supported by the University of Cambridge BBSRC Doctoral Training Programme (BB/M011194/1). CK was supported by European Commission Marie Skłodowska-Curie Fellowship MENTALIZINGORIGINS (Grant reference: 752373). NSC was funded by the European Research Council under the European Union's Seventh Framework Programme (FP7/2007-2013)/ERC Grant Agreement No. 3399933, awarded to NSC, and provided financial support conducting this research.

## Additional information

### Funding

| Funder | Grant reference number | Author |
|---|---|---|
| Leverhulme Trust | Study Abroad Scholarship SAS-2020-004\10 | Piero Amodio |
| Biotechnology and Biological Sciences Research Council | Doctoral Training Programme BB/M011194/1 | Benjamin G Farrar |
| European Commission | Marie Skłodowska-Curie Fellowship MENTALIZINGORIGINS Grant reference: 752373 | Christopher Krupenye |
| FP7 Ideas: European Research Council | ERC Grant Agreement N 3399933 | Nicola S Clayton |

The funders had no role in study design, data collection and interpretation, or the decision to submit the work for publication.

### Author contributions

Piero Amodio, Conceptualization, Data curation, Formal analysis, Investigation, Methodology, Visualization, Writing - original draft, Writing – review and editing; Benjamin G Farrar, Conceptualization, Investigation, Methodology, Writing – review and editing; Christopher Krupenye, Conceptualization, Methodology, Writing – review and editing; Ljerka Ostojić, Conceptualization, Investigation, Methodology, Supervision, Writing – review and editing; Nicola S Clayton, Funding acquisition, Supervision, Writing – review and editing

### Author ORCIDs

Piero Amodio http://orcid.org/0000-0002-9408-2902
Christopher Krupenye http://orcid.org/0000-0003-2029-1872
Ljerka Ostojić http://orcid.org/0000-0002-8008-1773
Nicola S Clayton http://orcid.org/0000-0003-1835-423X

## Ethics

All procedures were approved by the University of Cambridge Animal Ethics Committee (reference n. ZOO35/17).

## Decision letter and Author response

Decision letter https://doi.org/10.7554/eLife.69647.sa1
Author response https://doi.org/10.7554/eLife.69647.sa2

## Additional files

### Supplementary files

• Transparent reporting form

### Data availability

Data and analyses of all experiments are available at http://doi.org/10.5281/zenodo.4636561.

The following dataset was generated:

| Author(s) | Year | Dataset title | Dataset URL | Database and Identifier |
|---|---|---|---|---|
| Amodio P, Farrar BG, Krupenye C, Ostojic L, Clayton NS | 2021 | Little evidence that Eurasian jays (Garrulus glandarius) protect their caches by responding to cues about a conspecific's desire and visual perspective | http://doi.org/10.5281/zenodo.4636561 | Zenodo, 10.5281/zenodo.4636561 |

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
