## [Decision Letter]

**Acceptance summary:**

This paper aims to replicate influential findings that imply 'Theory of Mind' in food-caching decisions of Eurasian Jays. The authors' attempts to expand on and replicate earlier findings are both rigorous and thoroughly contextualized. The failure to reproduce earlier findings raises important questions for the field. The authors are to be commended for their unbiased, transparent and open-minded study.

**Decision letter after peer review:**

Thank you for submitting your article "Little evidence that Eurasian jays protect their caches by responding to cues about a conspecific's desire and visual perspective" for consideration by *eLife*. Your article has been reviewed by 2 peer reviewers, and the evaluation has been overseen by a Reviewing Editor and Detlef Weigel as the Senior Editor. The reviewers have opted to remain anonymous.

Essential revisions:

Please pay attention to the recommendations from the reviewers.

Reviewer #2 (Recommendations for the authors):

– More information should be added concerning what the implications would be for the field of avian cognition if the null findings observed here wind up holding up.

–The description of Experiment 4 in the text should make some reference to the fact that Legg and Clayton (2014) manipulated the dominance of the observer. I had forgotten this fact about this paper and so it was confusing when the dominant and subordinate observer conditions are shown in the schematic of Study 4.

-Figure 4's schematic of Study 5 includes the subordinate observer icon in the legend, but it doesn't actually appear in the figure, so it should be removed.

-Lines 480-482. This sentence that starts on line 480 is very hard to parse. I would suggest rewriting it to remove the passive voice.

-Lines 508-510. This sentence is stating the key findings of experiment 1, but it is a bit hard to understand since there are two findings. I suggest breaking it up into two separate sentences.

-Line 578. There is a space missing between retrieval and the Clayton et al., 2005 citation.

-Line 579. There is an extra space after in citation and before the period.

-Line 585 suggests that even the original tests may have had low validity since caching motivation may decrease after initial exposure to the novel set-up. Thus, it would be informative to include a brief description of how much exposure the birds had to these set-ups prior to Legg and Clayton, 2014 and Ostojic et al., 2017.

-Line 629. Define reliability test. While most researchers are probably familiar with the concept, they may not be familiar with that particular term.

---

## [Author Response]

Essential Revisions (for the authors):Please pay attention to the recommendations from the reviewers.Reviewer #2 (Recommendations for the authors):– More information should be added concerning what the implications would be for the field of avian cognition if the null findings observed here wind up holding up.

We have carefully taken this important point on board. Please refer toour comment in response to the similar issue raised by the reviewer in their Public Review, for a reply to this point.

–The description of Experiment 4 in the text should make some reference to the fact that Legg and Clayton (2014) manipulated the dominance of the observer. I had forgotten this fact about this paper and so it was confusing when the dominant and subordinate observer conditions are shown in the schematic of Study 4.

Thank you for pointing out this oversight. We have added a sentence (see lines 416-418 of the revised manuscript) to remind the reader that in Experiment 4 (as in Legg and Clayton 2014) jays were tested in three conditions: Observed by Dominant, Observed by Subordinate, and in Private.

-Figure 4's schematic of Study 5 includes the subordinate observer icon in the legend, but it doesn't actually appear in the figure, so it should be removed.

We thank the reviewer for spotting this mistake. We have replaced the ‘Subordinate observer’ icon (as well as the ‘Dominant observer’ icon) with the ‘Observer’ icon in the legend of Figure 4.

-Lines 480-482. This sentence that starts on line 480 is very hard to parse. I would suggest rewriting it to remove the passive voice.

We have adjusted the sentence according to the reviewer’s suggestion (see lines 485-489 of the revised manuscript).

-Lines 508-510. This sentence is stating the key findings of experiment 1, but it is a bit hard to understand since there are two findings. I suggest breaking it up into two separate sentences.

We have followed the suggestion and split up this sentence into two (see lines 521-529 of the revised manuscript). For consistency, we have also split the subsequent sentence in which we report the key findings of Experiment 2 into two.

-Line 578. There is a space missing between retrieval and the Clayton et al., 2005 citation.-Line 579. There is an extra space after in citation and before the period.

We thank the reviewer for spotting the two typos. Both errors have been corrected.

-Line 585 suggests that even the original tests may have had low validity since caching motivation may decrease after initial exposure to the novel set-up. Thus, it would be informative to include a brief description of how much exposure the birds had to these set-ups prior to Legg and Clayton, 2014 and Ostojic et al., 2017.

The ‘original tests’ in this sentence referred to our Experiments 1 and 2, not to the original experiments as in the Legg and Clayton (2014) and Ostojic et al., (2017) studies. We realise this wording is misleading and have edited this in the revised manuscript (see lines 630-632). What we referred to here is not the very general context of caching with an observer present (which the jays also experience with in the aviary) but the very specific set-ups that facilitate cache-protection strategies in question, for example the presence of a T-shaped barrier and two caching trays, caching specific foods after having watched another jay (the later observer) becoming sated on one of the two foods, etc. It is in these specific settings where jays may rapidly learn that caches are not pilfered and cache protection strategies are not necessary.

-Line 629. Define reliability test. While most researchers are probably familiar with the concept, they may not be familiar with that particular term.

We have adjusted the sentence to clarify what we meant by ‘reliability test’ (see lines 721-728 of the revised manuscript).